# Statistical Parity with Exponential Weights

**Stephen Pasteris**
The Alan Turing Institute
London UK
spasteris@turing.ac.uk

**Chris Hicks**
The Alan Turing Institute
London UK
c.hicks@turing.ac.uk

**Vasilios Mavroudis**
The Alan Turing Institute
London UK
vmavroudis@turing.ac.uk

## Abstract

Statistical parity is one of the most foundational constraints in algorithmic fairness and privacy. In this paper, we show that statistical parity can be enforced efficiently in the adversarial contextual bandit setting while retaining strong performance guarantees. Specifically, we present a meta-algorithm that transforms any efficient implementation of Hedge (or, equivalently, any discrete Bayesian inference algorithm) into an efficient contextual bandit algorithm that guarantees exact statistical parity on every trial. Compared to any comparator that satisfies the same statistical parity constraint, the algorithm achieves the same asymptotic regret bound as running the equivalent instance of Exp4 for each group. We also address the scenario where the target parity distribution is unknown and must be estimated online. Finally, using online-to-batch conversion, we extend our approach to the batch classification setting.

## 1 Introduction

*Statistical parity* [12] is a foundational concept in algorithmic fairness and privacy. It imposes a constraint on how decisions should be distributed across individuals with different values of a *protected characteristic*. Formally, let $\mathcal{C}$ and $\mathcal{V}$ denote the sets of protected characteristics and non-protected features (known as *contexts*) respectively, and let $\rho$ be a probability distribution over $\mathcal{C} \times \mathcal{V}$. A *policy* mapping $\mathcal{C} \times \mathcal{V}$ to distributions over a set of *actions* is said to satisfy statistical parity with respect to $\rho$ if and only if, when a pair $(c, v)$ is drawn from $\rho$ and then an action $a$ is sampled from the policy, the selected action $a$ is independent of the protected characteristic $c$. That is, for all actions $b$ and all protected characteristics $d, d' \in \mathcal{C}$ we have:

$$\mathbb{P}[a = b \mid c = d] = \mathbb{P}[a = b \mid c = d']$$

We now provide two illustrative examples motivating the enforcement of statistical parity. The first concerns *stop-and-search* procedures, in which law enforcement officers select individuals from the public to search for prohibited items. This practice has been the subject of substantial controversy due to evidence of racial bias - with individuals from certain ethnic backgrounds being many times more likely to be stopped than others. In this setting, there are two possible actions: *stop* and *do not stop*, and the protected characteristic corresponds to an individual's ethnicity. Enforcing statistical parity in this context ensures that the probability of being stopped is equal across all ethnic groups, thereby eliminating disparities attributable to racial bias.

The second example arises in the context of cyber defence. Consider a computer system that must be protected using a defensive strategy, often referred to as a *blue agent*. The choice of blue agent depends on certain private attributes of the system, one of which is particularly sensitive. Through interaction with the system, external users may be able to infer which blue agent was selected, thereby gaining information about the underlying attributes. In this case, the actions correspond to the chosen blue agents, and the protected characteristics are the possible values of the sensitive attribute. Enforcing statistical parity ensures that the selected blue agent is independent of the sensitive attribute, thereby preventing users from inferring anything about it.

39th Conference on Neural Information Processing Systems (NeurIPS 2025).

In this paper, we primarily study the *contextual bandit problem*, where learning unfolds over a sequence of *trials* in an online setting. On each trial, the learner selects a policy, then observes an *instance* from the set $\mathcal{C} \times \mathcal{V}$, and samples an action according to the chosen policy. After the action is taken, the learner receives the *loss* (or, equivalently, a reward) corresponding to that action on that trial. The goal is to minimise the total cumulative loss incurred over all trials. The algorithm's performance is measured via *regret*, defined as the difference between the expected cumulative loss of the algorithm and that of a fixed comparator policy (or, in some cases, a sequence of comparator policies) in hindsight. We focus mainly on the *adversarial* setting, in which the only assumption is that the losses lie in $[0, 1]$, and there are no restrictions on how instances or losses are generated. Additionally, we consider cases where the context space exhibits structure, introducing inductive biases to exploit this structure.

To the best of our knowledge, no prior work has successfully developed an efficient algorithm for the contextual bandit problem that ensures statistical parity in the selected policies, even in the simplified setting of a finite, unstructured context space with i.i.d. instances and losses. This paper addresses that gap. We introduce SPEW (Statistical Parity with Exponential Weights), an efficient algorithm that achieves statistical parity while matching the asymptotic regret guarantees of the classic EXP4 algorithm [2]. Like EXP4, SPEW is a meta-algorithm that leverages any efficient instantiation of HEDGE [13] adapted to the structure of the context space.

As an example, we analyse the regret bound of SPEW for when the contexts are the set of vertices of a tree (noting that graphs and finite metric spaces reduce to trees). In addition to guaranteeing statistical parity with respect to a given parity distribution, SPEW also extends to settings where the parity distribution is unknown and must be estimated from the data seen so far. Specifically, we show how to enforce statistical parity with respect to the empirical distribution of observed instances, as well as how to maintain statistical parity approximately when the context space is hierarchically clustered (such as graphs or Euclidean spaces) and the instances are drawn i.i.d. from the parity distribution.

Beyond the contextual bandit setting, SPEW can also be applied to the batch classification problem under statistical parity constraints via standard online-to-batch conversion techniques [9] . We believe this yields novel results for the batch fairness and privacy literature. Moreover, the underlying methodology of SPEW is not necessarily limited to HEDGE-based implementations; it should extend naturally to other exponentiated gradient algorithms, and potentially even to broader classes of gradient-based methods. For instance, our approach should be able to adapt to the CBA algorithm [27], enabling the incorporation of confidence-rated expert advice, which further broadens the applicability of our framework.

We now explain why a direct modification of EXP4 to incorporate statistical parity constraints is problematic. EXP4 modifies the inputs to HEDGE: an algorithm that performs mirror descent over the probability simplex of experts (i.e. deterministic policies). In principle, one could attempt to incorporate statistical parity by adding the constraint directly into the mirror descent framework. This would require performing a relative entropy projection onto the constrained set. To compute such a projection, we construct the Lagrangian and differentiate it to obtain necessary conditions for optimality. While expressing the primal variables in terms of the Lagrange multipliers is straightforward, solving for the multipliers themselves appears analytically intractable. Although the projection problem is convex and thus admits numerical solutions, known approximation methods are, as far as we are aware, computationally infeasible in this context due to the exponential size of the expert space. Even in the special case where there is no structure to the context set - effectively removing the inductive bias and reducing the problem to polynomial dimension - existing approximation methods remain considerably less efficient than our approach. In contrast, SPEW avoids these issues entirely by not modifying HEDGE itself but rather its inputs and outputs.

Much research has focused on designing efficient algorithms for online convex optimization that, unlike SPEW, allow constraint violations but ensure bounded cumulative constraint violation. Notably, the work [8] extends to the bandit setting and achieves a cumulative constraint violation of $\mathcal{O}(T^{3/4})$ (suppressing factors independent of $T$), albeit at the cost of an additional multiplicative term of $\mathcal{O}(M^{1/2}T^{1/4})$ to the regret. The work [34] attains a tighter constraint violation bound of $\mathcal{O}(T^{1/2})$ (when the constraint set lacks an interior point, as for statistical parity) but their approach is incompatible with the bandit setting and incurs polynomial dependence on the problem dimension (which is exponential in our work) in both the regret and constraint violation bounds.

We provide a literature review in Appendix A.

## 2  Notation

In this section we introduce the notation used in this paper. We define $\mathbb{R}_+$ to be the set of non-negative real numbers. We define $\mathbb{N}$ to be the set of natural numbers excluding $0$. Given $z \in \mathbb{N}$ we define:

$$[z] := \{z' \in \mathbb{N} \,|\, z' \leq z\}$$

Given some finite set $\mathcal{B}$ we define $\Delta_{\mathcal{B}}$ to be the set of probability distributions over $\mathcal{B}$. That is, $\Delta_{\mathcal{B}}$ is the set of all $\rho : \mathcal{B} \to [0,1]$ with:

$$\sum_{z \in \mathcal{B}} \rho(z) = 1$$

We call the set $\Delta_{\mathcal{B}}$ a *simplex*. Given sets $\mathcal{B}$ and $\mathcal{B}'$ and a function $f$ with domain $\mathcal{B} \times \mathcal{B}'$ we define, for all $z \in \mathcal{B}$, the function $f(z, \circ)$ to be the function that maps each $z' \in \mathcal{B}'$ to $f(z, z')$. Given a predicate $P$ we define $[\![P]\!] := 1$ if $P$ is true and $[\![P]\!] := 0$ otherwise.

## 3  Problem Description

In this paper we consider the following problem, known as the *contextual bandit problem*. We have a set of *contexts* $\mathcal{V}$ which typically has some structure associated with it. For instance, $\mathcal{V}$ could be the set of vertices of a graph or points in Euclidean space. When $\mathcal{V}$ is finite we define $N := |\mathcal{V}|$. We also have finite sets $\mathcal{C}$ and $\mathcal{A}$ of *protected characteristics* and *actions* respectively. Let $M := |\mathcal{C}|$ and $K := |\mathcal{A}|$. Define the set of *policies* as:

$$\mathcal{Q} := \left\{ \pi \in [0,1]^{\mathcal{C} \times \mathcal{V} \times \mathcal{A}} \,|\, \forall (c,v) \in \mathcal{C} \times \mathcal{V}, \, \pi(c,v,\circ) \in \Delta_{\mathcal{A}} \right\}$$

A-priori, our adversary chooses a sequence:

$$\langle (c_t, v_t, \ell_t) \,|\, t \in [T] \rangle$$

where for all $t \in [T]$ we have $c_t \in \mathcal{C}$, $v_t \in \mathcal{V}$ and $\ell_t \in [0,1]^{\mathcal{A}}$. This sequence is not revealed to us. The problem proceeds in $T$ *trials* where on trial $t$ the following happens.

1. We implicitly select a policy $\pi_t^* \in \mathcal{Q}$.
2. The pair $(c_t, v_t)$ is revealed to us.
3. We draw an action $a_t$ from $\pi_t^*(c_t, v_t, \circ)$.
4. The *loss* $\ell_t(a_t)$ is revealed to us.

Our aim is to minimise the *cumulative loss*:

$$\sum_{t \in [T]} \ell_t(a_t)$$

In this paper we consider the enforcement of *statistical parity* in our policy selections. Formally, a policy $\pi \in \mathcal{Q}$ has statistical parity with respect to a probability distribution $\rho \in \Delta_{\mathcal{C} \times \mathcal{V}}$ if and only if when a pair $(c,v)$ is drawn from $\rho$ and then an action $a$ is drawn from $\pi(c,v,\circ)$ we have that:

$$\mathbb{P}[a = b \,|\, c = d] = \mathbb{P}[a = b \,|\, c = d']$$

for all $b \in \mathcal{A}$ and $d, d' \in \mathcal{C}$. In other words, the selected action $a$ is independent of the protected characteristic $c$. Note that when $\mathbb{P}[c = d] = 0$ then we allow $\mathbb{P}[a = b \,|\, c = d]$ to be arbitrary in the above equation.

Given we have knowledge of a distribution $\rho \in \Delta_{\mathcal{C} \times \mathcal{V}}$, our meta-algorithm SPEW will play in such a way that each policy $\pi_t^*$ has statistical parity with respect to $\rho$. We note that the contexts $v_t$ selected by our adversary need not be drawn from $\rho$. Specifically, given any efficient implementation of an instance of the classic HEDGE algorithm, SPEW will convert it into an efficient algorithm that will guarantee statistical parity whilst also guaranteeing that the bound on the difference (a.k.a. the *regret*) between its cumulative loss and that which would have been incurred by any constant policy with statistical parity is that of the equivalent instance of EXP4.

As we shall see, SPEW can also work when we have no knowledge of $\rho$, and instead must estimate it from the data seen so far. SPEW can also be applied to cases in which we want to compare against a dynamic policy sequence rather than a constant policy.

# 4 The Algorithm

## 4.1 Hedge

Here we describe how an instance of HEDGE works. We have a finite set $\mathcal{X}$ of *virtual contexts* which is defined by the specific instance of HEDGE. On each trial $t$ we have a non-empty set $\mathcal{G}'_t \subseteq \mathcal{X}$ of the virtual contexts that are *relevant* for that trial, as well as a function $\chi'_t : \mathcal{V} \to \mathcal{G}'_t$ that will map, on trial $t$, the given context $v_t$ to a virtual context. These mathematical objects, which are defined by the specific instance of HEDGE, may be dependent on characteristic/context pairs seen before trial $t$. In many cases we will have $\mathcal{X} = \mathcal{V}$, $\mathcal{G}'_t = \mathcal{X}$ and $\chi'_t(v) = v$ for all $t \in [T]$ and $v \in \mathcal{V}$. We also have a *learning rate* $\hat{\eta} > 0$.

We define the set of *experts* as:

$$\mathcal{H} := \mathcal{A}^{\mathcal{X}}$$

so that each expert associates an action with each virtual context. The inductive bias of the specific instance of HEDGE is given by a probability distribution $\vartheta \in \Delta_{\mathcal{H}}$ over the experts. HEDGE implicitly maintains a distribution $\vartheta' \in \Delta_{\mathcal{H}}$ initialised equal to $\vartheta$. HEDGE has the following two subroutines:

- When run on trial $t$ the subroutine QUERY returns a function $\varphi : \mathcal{G}'_t \times \mathcal{A} \to [0, 1]$ defined by:
$$\varphi(x, a) := \sum_{e \in \mathcal{H}} [\![e(x) = a]\!] \vartheta'(e) \quad \forall (x, a) \in \mathcal{G}'_t \times \mathcal{A}$$

- When run on trial $t$ the subroutine UPDATE takes, as input, a pair $(x, \ell) \in \mathcal{G}'_t \times \mathbb{R}^{\mathcal{A}}$ and then, for all $e \in \mathcal{H}$, implicitly sets:
$$\vartheta'(e) \leftarrow \vartheta'(e) \exp(-\hat{\eta}\ell(e(x)))$$
and finally implicitly normalises $\vartheta'$.

Note that if HEDGE was run explicitly then these subroutines would take exponential time. However, given a compatible inductive bias $\vartheta$, algorithms for Bayesian inference (such as BELIEFPROPAGATION [29]) can compute these subroutines efficiently. Such algorithms typically require only $\mathcal{O}(K)$ time for the UPDATE subroutine as the main computation is performed during QUERY.

SPEW maintains an instance of HEDGE (a.k.a. the *base algorithm*) for each protected characteristic. For simplicity, we assume each of these instances of HEDGE shares the same set $\mathcal{X}$ and inductive bias $\vartheta$ although this is not actually necessary. For each $c \in \mathcal{C}$ and $t \in [T]$ define $\mathcal{G}_t(c)$ and $\chi_t(c, \circ)$ to be the mathematical objects $\mathcal{G}'_t$ and $\chi'_t$ for the instance of HEDGE associated with protected characteristic $c$. In addition, let QUERY$[c]$ and UPDATE$[c]$ be the subroutines QUERY and UPDATE for the instance of HEDGE associated with protected characteristic $c$.

## 4.2 Input and output

We now describe the input to SPEW, on each trial $t \in [T]$, and the properties of the action $a_t$ that it selects. To do this we first define the *virtual policy* space:

$$\mathcal{P} := \left\{ \pi \in [0, 1]^{\mathcal{C} \times \mathcal{X} \times \mathcal{A}} \,|\, \forall (c, x) \in \mathcal{C} \times \mathcal{X} \,,\, \pi(c, x, \circ) \in \Delta_{\mathcal{A}} \right\}$$

and the *target* space:

$$\mathcal{D} := \left\{ \mu \in [0, 1]^{\mathcal{C} \times \mathcal{X}} \,|\, \forall c \in \mathcal{C} \,,\, \mu(c, \circ) \in \Delta_{\mathcal{X}} \right\}$$

Finally, for any target $\mu \in \mathcal{D}$, we define $\mathcal{F}(\mu)$ to be the set of all virtual policies $\pi \in \mathcal{P}$ in which for all $c, c' \in \mathcal{C}$ and $a \in \mathcal{A}$ we have:

$$\sum_{x \in \mathcal{X}} \mu(c, x)\pi(c, x, a) = \sum_{x \in \mathcal{X}} \mu(c', x)\pi(c', x, a)$$

On each trial $t \in [T]$ the following happens:

1. A target $\mu_t \in \mathcal{D}$ is created based on the characteristic/context pairs seen so far and the problem at hand. This target has the property that for all $(c, x) \in \mathcal{C} \times \mathcal{X}$ with $x \notin \mathcal{G}_t(c)$ we have $\mu_t(c, x) = 0$.

2. SPEW implicitly determines a virtual policy $\pi_t \in \mathcal{F}(\mu_t)$. This virtual policy equates to the policy $\pi_t^* \in \mathcal{Q}$ by $\pi_t^*(c, v, \circ) := \pi_t(c, \chi_t(c, v), \circ)$ for all $(c, v) \in \mathcal{C} \times \mathcal{V}$.

3. $(c_t, v_t)$ is revealed.

4. We define $x_t := \chi_t(c_t, v_t)$.

5. SPEW samples the action $a_t$ from the distribution $\pi_t(c_t, x_t, \circ)$. Note that this means that SPEW samples $a_t$ from $\pi_t^*(c_t, v_t, \circ)$ as required.

6. $\ell_t(a_t)$ is revealed.

We now describe how this process enforces statistical parity. Specifically, suppose we have a distribution $\rho \in \Delta_{\mathcal{C} \times \mathcal{V}}$. We assume, for simplicity, that for each $c \in \mathcal{C}$ there exists some $x \in \mathcal{X}$ with $\rho(c, x) \neq 0$ although it is trivial to modify the algorithm so that it does not require such an assumption. On any trial $t$, we can define $\mu_t$ by:

$$\mu_t(c, x) := \frac{\sum_{v \in \mathcal{V}} [\![\chi_t(c, v) = x]\!] \rho(c, v)}{\sum_{v \in \mathcal{V}} \rho(c, v)} \quad \forall (c, x) \in \mathcal{C} \times \mathcal{X}$$

Note that for all $(c, x) \in \mathcal{C} \times \mathcal{X}$ with $x \notin \mathcal{G}_t(c)$ we have $\mu_t(c, x) = 0$ as required. Since $\pi_t \in \mathcal{F}(\mu_t)$ we then have that the policy $\pi_t^*$ has statistical parity with respect to $\rho$.

### 4.3 Pseudocode

We now give the pseudocode of SPEW. On any trial $t \in [T]$ SPEW does the following.

Receive $\mu_t, x_t$ and $c_t$
**for** $c \in \mathcal{C}$ **do**
  $\xi_t(c, \circ, \circ) \leftarrow \text{QUERY}[c]$
  **for** $a \in \mathcal{A}$ **do**
    $\omega_t(c, a) \leftarrow \sum_{x \in \mathcal{G}_t(c)} \mu_t(c, x) \xi_t(c, x, a)$
  **end for**
**end for**
**for** $(c, a) \in \mathcal{C} \times \mathcal{A}$ **do**
  $\delta_t(c, a) \leftarrow \max_{c' \in \mathcal{C}} \omega_t(c', a) - \omega_t(c, a)$
**end for**
$\beta_t \leftarrow \sum_{a \in \mathcal{A}} \max_{c \in \mathcal{C}} \delta_t(c, a)$
**for** $a \in \mathcal{A}$ **do**
  $\psi_t'(a) \leftarrow (\xi_t(c_t, x_t, a) + \delta_t(c_t, a))/(1 + \beta_t)$
**end for**
$h_t \leftarrow 1 - \sum_{a \in \mathcal{A}} \psi_t'(a)$
**for** $a \in \mathcal{A}$ **do**
  $\pi_t'(a) \leftarrow \psi_t'(a) + h_t/K$
**end for**
Draw $a_t$ from probability distribution $\pi_t'$
Receive $\ell_t(a_t)$
**for** $a \in \mathcal{A}$ **do**
  $\kappa_t(a) \leftarrow \text{argmax}_{c \in \mathcal{C}} \omega_t(c, a)$
  $\kappa_t'(a) \leftarrow \text{argmin}_{c \in \mathcal{C}} \omega_t(c, a)$
**end for**
**for** $c \in \mathcal{C}$ **do**
  **for** $x \in \mathcal{G}_t(c)$ **do**
    **for** $a \in \mathcal{A}$ **do**
      $\lambda_t(c, x, a) \leftarrow [\![\beta_t \leq 1]\!][\![(c, x, a) = (c_t, x_t, a_t)]\!]\ell_t(a_t)/\pi_t'(a_t)$
        $+ [\![c = \kappa_t(a)]\!]\mu_t(c, x) - [\![c = \kappa_t'(a)]\!]\mu_t(c, x)$
    **end for**
    $\text{UPDATE}[c](x, \lambda_t(c, x, \circ))$
  **end for**
**end for**

## 4.4 Description

We now briefly describe the internal workings of SPEW on trial $t$. For a full description and analysis please see Appendix B. On a high level, SPEW maintains a virtual policy (which does not necessarily satisfy statistical parity) inside the instances of HEDGE. This virtual policy is computed by calling the QUERY subroutine of each instance of HEDGE. On trial $t$, this virtual policy $\xi_t$ is mapped, in a certain way, to the virtual policy $\pi_t$, which does satisfy statistical parity. $\xi_t$ is then updated to $\xi_{t+1}$ by performing exponentiated gradient descent (on the expert probabilities) using an unbiased estimate of the gradient of a convex surrogate of the loss of $\pi_t$ in terms of $\xi_t$. This exponentiated gradient descent update is performed by updating the instances of HEDGE (via the UPDATE subroutine) multiple times.

We now go into details. Given some $c \in \mathcal{C}$, recall (from Section 4.1) that the instance of the HEDGE for protected characteristic $c$ implicitly maintains a distribution in $\Delta_{\mathcal{H}}$. Let $\vartheta_t(c, \circ)$ be the value of this distribution at the start of trial $t$, so that $\vartheta_1(c, \circ) = \vartheta$.

Define the virtual policy $\xi_t \in \mathcal{P}$ by:

$$\xi_t(c, x, a) := \sum_{e \in \mathcal{H}} [\![ e(x) = a ]\!] \vartheta_t(c, e)$$

for all $(c, x, a) \in \mathcal{C} \times \mathcal{X} \times \mathcal{A}$. We call $\xi_t$ the *raw* policy. For each $c \in \mathcal{C}$ SPEW needs only know $\xi_t(c, x, \circ)$ for $x \in \mathcal{G}_t(c)$, which it can compute by a call to QUERY$[c]$.

Note that the raw policy does not necessarily belong to $\mathcal{F}(\mu_t)$. The raw policy is converted into the virtual policy $\pi_t$ by the following procedure. Given $(c, a) \in \mathcal{C} \times \mathcal{A}$ define:

$$\omega_t(c, a) := \sum_{x \in \mathcal{G}_t(c)} \mu_t(c, x) \xi_t(c, x, a)$$

and:

$$\delta_t(c, a) := \max_{c' \in \mathcal{C}} \omega_t(c', a) - \omega_t(c, a)$$

Furthermore define:

$$\beta_t := \sum_{a \in \mathcal{A}} \max_{c \in \mathcal{C}} \delta_t(c, a)$$

For all $(c, x, a) \in \mathcal{C} \times \mathcal{X} \times \mathcal{A}$ define:

$$\psi_t(c, x, a) := \frac{\xi_t(c, x, a) + \delta_t(c, a)}{1 + \beta_t}$$

Finally, define the virtual policy $\pi_t \in \mathcal{P}$ by:

$$\pi_t(c, x, a) := \psi_t(c, x, a) + \frac{1}{K} \left( 1 - \sum_{a' \in \mathcal{A}} \psi_t(c, x, a') \right)$$

for all $(c, x, a) \in \mathcal{C} \times \mathcal{X} \times \mathcal{A}$. In Appendix B we show that $\pi_t \in \mathcal{F}(\mu_t)$ as required.

SPEW then samples the action $a_t$ from the distribution $\pi_t(c_t, x_t, \circ)$.

Now that we have shown how to choose the action $a_t$, we will show how to update the HEDGE instances upon receipt of $\ell_t(a_t)$. We first define:

$$\mathcal{S} := \mathbb{R}^{\mathcal{C} \times \mathcal{X} \times \mathcal{A}}$$

and given $(\phi, c, a) \in \mathcal{S} \times \mathcal{C} \times \mathcal{A}$ define:

$$\tilde{\omega}_t(\phi, c, a) := \sum_{x \in \mathcal{G}_t(c)} \mu_t(c, x) \phi(c, x, a)$$

We define the function $y_t : \mathcal{S} \to \mathbb{R}$ by:

$$y_t(\phi) := [\![ \beta_t \leq 1 ]\!] \sum_{a \in \mathcal{A}} \ell_t(a) \phi(c_t, x_t, a) + \sum_{a \in \mathcal{A}} \left( \max_{c \in \mathcal{C}} \tilde{\omega}_t(\phi, c, a) - \min_{c \in \mathcal{C}} \tilde{\omega}_t(\phi, c, a) \right)$$

noting that SPEW never actually has knowledge of this function, as it is dependent on the entirety of $\ell_t$. In Appendix B we show that $y_t$ is convex and that:

$$y_t(\xi_t) \geq \sum_{a \in \mathcal{A}} \pi_t(c_t, x_t, a) \ell_t(a) \quad ; \quad y_t(\tilde{\pi}) \leq \sum_{a \in \mathcal{A}} \tilde{\pi}(c_t, x_t, a) \ell_t(a) \quad \forall \tilde{\pi} \in \mathcal{F}(\mu_t)$$

These properties mean that we can use $y_t$ as a convex surrogate for the expected loss in terms of the raw policy.

Since SPEW does not know the entirety of $\ell_t$ it does not know a subgradient of $y_t$ at $\xi_t$ - which is required for exponentiated gradient descent. Hence, we borrow the idea from EXP4, of constructing an unbiased estimate of such a subgradient. Specifically, we define the function $\lambda_t \in \mathcal{S}$ by:

$$\lambda_t(c, x, a) := [\![ \beta_t \leq 1 ]\!] [\![ (c, x, a) = (c_t, x_t, a_t) ]\!] \frac{\ell_t(a_t)}{\pi_t(c_t, x_t, a_t)}$$
$$+ [\![ c = \mathrm{argmax}_{c' \in \mathcal{C}} \, \omega_t(c', a) ]\!] \mu_t(c, x) - [\![ c = \mathrm{argmin}_{c' \in \mathcal{C}} \, \omega_t(c', a) ]\!] \mu_t(c, x)$$

where ties in the argmax and argmin are broken arbitrarily. In Appendix B we prove that the expected value of $\lambda_t$ over the draw of $a_t$ is equal to a subgradient of $y_t$ at $\xi_t$. Noting that, for each $(c, e) \in \mathcal{C} \times \mathcal{H}$, $\xi_t$ is dependent on $\vartheta_t(c, e)$, we then have that an unbiased estimate of the subgradient of $y_t(\xi_t)$ with respect to $\vartheta_t(c, e)$ is equal to:

$$\nu_t(c, e) := \sum_{x \in \mathcal{X}} \lambda_t(c, x, e(x))$$

We can now update the function $\vartheta_t$ to $\vartheta_{t+1}$ via exponentiated gradient descent. Specifically, for all $(c, e) \in \mathcal{C} \times \mathcal{H}$ we have:

$$\vartheta_{t+1}(c, e) := \frac{\vartheta_t(c, e) \exp(-\hat{\eta} \nu_t(c, e))}{\sum_{e' \in \mathcal{H}} \vartheta_t(c, e') \exp(-\hat{\eta} \nu_t(c, e'))}$$

This updated is implemented by simply running the function

$$\mathrm{UPDATE}[c](x, \lambda_t(c, x, \circ))$$

for each $c \in \mathcal{C}$ and $x \in \mathcal{G}_t(c)$.

### 4.5 Performance

In the following theorem we give the general performance guarantee of SPEW.

**Theorem 4.1.** *Let:* $\eta := \hat{\eta}\sqrt{KT}$. *For all* $t \in [T]$, SPEW *produces* $\pi_t \in \mathcal{F}(\mu_t)$. *In addition, for any virtual policy:*

$$\tilde{\pi} \in \bigcap_{t \in [T]} \mathcal{F}(\mu_t)$$

*and for any* $\vartheta^* : \mathcal{C} \times \mathcal{H} \to [0, 1]$ *with:*

$$\tilde{\pi}(c, x, a) = \sum_{e \in \mathcal{H}} [\![ e(x) = a ]\!] \vartheta^*(c, e) \quad \forall (c, x, a) \in \mathcal{C} \times \mathcal{X} \times \mathcal{A}$$

*we have:*

$$\mathbb{E}\left[ \sum_{t \in [T]} \ell_t(a_t) \right] \leq \sum_{t \in [T]} \sum_{a \in \mathcal{A}} \tilde{\pi}(c_t, x_t, a) \ell_t(a) + \mathcal{O}\left( \left( \eta + \frac{\Phi}{\eta} \right) \sqrt{KT} \right)$$

*where:*

$$\Phi := \sum_{c \in \mathcal{C}} \sum_{e \in \mathcal{H}} \vartheta^*(c, e) \ln \left( \frac{\vartheta^*(c, e)}{\vartheta(e)} \right)$$

*On each trial* $t \in [T]$ *the time complexity of* SPEW *is that of* $M$ *calls to* QUERY *and* $\sum_{c \in \mathcal{C}} |\mathcal{G}_t(c)|$ *calls to* UPDATE. *The space complexity is that of maintaining* $M$ *instances of* HEDGE.

See Appendix B for the proof of this theorem. We note that for all applications the targets $\mu_t$ will be chosen such that $\bigcap_{t \in [T]} \mathcal{F}(\mu_t)$ is non-empty. We also note that for all virtual policies $\tilde{\pi} \in \mathcal{P}$ there exists $\vartheta^* : \mathcal{C} \times \mathcal{H} \to [0, 1]$ with:

$$\pi(c, x, a) = \sum_{e \in \mathcal{H}} [\![e(x) = a]\!] \vartheta^*(c, e) \quad \forall (c, x, a) \in \mathcal{C} \times \mathcal{X} \times \mathcal{A}$$

as is required by the theorem. Relative to any virtual policy $\tilde{\pi} \in \bigcap_{t \in [T]} \mathcal{F}(\mu_t)$, the regret bound given by the theorem is asymptotically identical to that of running EXP4 independently for each protected characteristic.

## 5 Examples

Before we discuss the algorithm and the main result, we will give some example applications. The specific instances of SPEW that are used to obtain the results of this section, as well as their proofs, are given in Appendix D. In the first example we assume that the distribution with which we must have statistical parity with respect to is given a-priori, whilst in the final two examples it is instead estimated from the data seen so far. Note that, in the theorems, the quantity:

$$\sum_{a \in \mathcal{A}} \tilde{\pi}(c_t, v_t, a) \ell_t(a)$$

is the expected loss incurred by playing policy $\tilde{\pi} \in \mathcal{Q}$ on trial $t \in [T]$. Note also that $\eta := \hat{\eta}\sqrt{KT}$

### 5.1 Trees with a known distribution

In this problem we assume we must maintain statistical parity with respect to a distribution $\rho \in \Delta_{\mathcal{C} \times \mathcal{V}}$ that is known to us a-priori. We also assume that we have knowledge of a tree with vertex set $\mathcal{V}$. Let $\mathcal{E}$ be the edge set of this tree. Several types of problem reduce to this setting, including learning on graphs [14], learning on finite metric spaces [26] and learning on finite sets endowed with a hierarchical clustering. For graphs, the tree is created by sampling a spanning tree uniformly at random. For metric spaces the tree is created by growing the tree context by context, linking each new context to it's nearest neighbour in the tree so far. For heirarchical clusterings the context set is altered to be the set of all clusters, which naturally has a tree structure.

Given a policy $\tilde{\pi} \in \mathcal{Q}$ we define its *complexity* as:

$$\Psi(\tilde{\pi}) := \sum_{(u,v) \in \mathcal{E}} \sum_{(c,a) \in \mathcal{C} \times \mathcal{A}} |\tilde{\pi}(c, u, a) - \tilde{\pi}(c, v, a)|$$

Note that $\Psi(\tilde{\pi})$ is bounded above by twice the summation, over all $c \in \mathcal{C}$, of the number of edges $(u, v) \in \mathcal{E}$ in which $\tilde{\pi}(c, u, \circ) \neq \tilde{\pi}(c, v, \circ)$ (i.e. the *cutsize*).

We have the following performance guarantee.

**Theorem 5.1.** *Suppose we are given a distribution $\rho \in \Delta_{\mathcal{C} \times \mathcal{V}}$ a-priori and run the appropriate instance of* SPEW. *Then all our generated policies $\pi_t^*$ have statistical parity with respect to $\rho$ and for any policy $\tilde{\pi} \in \mathcal{Q}$ with statistical parity with respect to $\rho$ we have:*

$$\mathbb{E}\left[\sum_{t \in [T]} \ell_t(a_t)\right] \leq \sum_{t \in [T]} \sum_{a \in \mathcal{A}} \tilde{\pi}(c_t, v_t, a)\ell_t(a) + \tilde{\mathcal{O}}\left(\left(\eta + \frac{M + \Psi(\tilde{\pi})\ln(N)}{\eta}\right)\sqrt{KT}\right)$$

*The per-trial time and space complexity is $\mathcal{O}(MNK)$.*

This theorem leads directly to theorems about learning on graphs and finite metric spaces via results from [10] (it is a commonly used fact that the expected cutsize of uniformly sampled spanning tree of a labelled graph is equal to the effective resistance weighted cutsize of the graph) and [26] (see Theorem 3.6 therein) respectively.

## 5.2 Empirical statistical parity

We now turn to examples in which we are not given a distribution $\rho$ a-priori. In this first example we utilise the methodology of [17] to enforce statistical parity with respect to the empirical distribution of the data seen so far. Note that for this to be meaningful the set $\mathcal{V}$ should not be too large (we will deal with very large and essentially infinite context sets in the next example). The set $\mathcal{V}$ could, for example, be created via a clustering of a larger context set. We do not exploit any structure to the contexts in this example.

For any trial $t > 1$ let $\rho_t$ be the empirical distribution of the data seen so far. Formally, for all $(c, v) \in \mathcal{C} \times \mathcal{V}$, we have:

$$\rho_t(c, v) := \frac{1}{t-1} \sum_{s \in [t-1]} [\![ c_s = c, v_s = v ]\!]$$

Given any sequence of policies $\tilde{\boldsymbol{\pi}} \in \mathcal{Q}^T$, we define its *complexity*:

$$\Psi(\tilde{\boldsymbol{\pi}}) := \sum_{t \in [T-1]} \sum_{(c,v,a) \in \mathcal{C} \times \mathcal{V} \times \mathcal{A}} |\tilde{\pi}_{t+1}(c, v, a) - \tilde{\pi}_t(c, v, a)|$$

With these definitions in hand we have the following result:

**Theorem 5.2.** *There exists an instance of* SPEW *such that for all $t > 1$ we have that $\pi_t^*$ has statistical parity with respect to $\rho_t$ and, for any sequence of policies $\tilde{\boldsymbol{\pi}} \in \mathcal{Q}^T$ in which, for all $t > 1$, $\tilde{\pi}_t$ has statistical parity with respect to $\rho_t$, we have that:*

$$\mathbb{E}\left[ \sum_{t \in [T]} \ell_t(a_t) \right] \leq \sum_{t \in [T]} \sum_{a \in \mathcal{A}} \tilde{\pi}_t(c_t, v_t, a)\ell_t(a) + \tilde{\mathcal{O}}\left( \left( \eta + \frac{MN + \Psi(\tilde{\boldsymbol{\pi}})}{\eta} \right) \sqrt{KT} \right)$$

*The per-trial time and space complexity is $\mathcal{O}(MNK)$.*

## 5.3 Approximate statistical parity for hierarchical clusterings with an unknown distribution

We now turn to maintaining approximate statistical parity when the context space is potentially massive and has a known hierarchical clustering associated with it. Natural hierarchical clusterings can be derived for several types of structure, including graphs [15], Euclidean space [30], and finite metric spaces [25].

This example, which utilises the methodology of [26], does not assume a-priori knowledge of a distribution $\rho$, but unlike the previous examples we assume here that there exists a distribution $\rho^* \in \Delta_{\mathcal{C} \times \mathcal{V}}$ in which the pairs $(c_t, v_t)$ are each drawn i.i.d. from $\rho^*$. This distribution is not known to us a-priori. We are given some $\epsilon, \delta > 0$. We say that a policy $\pi \in \mathcal{Q}$ has $\epsilon$-*approximate statistical parity* with respect to $\rho^*$ if and only if, when a pair $(c, v)$ is drawn from $\rho^*$ and then an action $a$ is drawn from $\pi(c, v, \circ)$, we have that:

$$\mathbb{P}[a = b \,|\, c = d'] - \epsilon \leq \mathbb{P}[a = b \,|\, c = d] \leq \mathbb{P}[a = b \,|\, c = d'] + \epsilon$$

for all $b \in \mathcal{A}$ and $d, d' \in \mathcal{C}$. We are required to play in such a way that, with probability at least $1 - \delta$, all our chosen policies $\pi_t^*$ have $\epsilon$-approximate statistical parity with respect to $\rho^*$.

We assume that $\mathcal{V}$ is finite but potentially enormous. For instance, $\mathcal{V}$ could be the quantisation of a hypercube of Euclidean space defined by all the points that the computer can handle (given its precision). We assume there is a known hierarchical clustering over $\mathcal{V}$. That is, we have a full binary tree in which each vertex is a subset of $\mathcal{V}$. The root is the whole set $\mathcal{V}$, the leaves are the singleton sets, and the two children of each internal vertex/set partition that set. We note that there is no need to explicitly maintain the tree (which could be enormous). We define $H$ to be the height of the hierarchical clustering, which is typically a relatively small quantity (e.g. logarithmic in $T$).

Given a policy $\tilde{\pi} \in \mathcal{Q}$ we define its *complexity* $\Psi(\tilde{\pi})$ as follows. We first define a *permitted clustering* to be any partition of $\mathcal{V}$ comprising of sets in the hierarchical clustering. For any protected characteristic $c \in \mathcal{C}$ we define $\psi(c, \tilde{\pi})$ to be the minimum cardinality of any permitted clustering in which for each set $\mathcal{U}$ in that permitted clustering and for all $u, v \in \mathcal{U}$ we have $\tilde{\pi}(c, u, \circ) = \tilde{\pi}(c, v, \circ)$. We then define:

$$\Psi(\tilde{\pi}) := \sum_{c \in \mathcal{C}} \psi(c, \tilde{\pi})$$

Finally, we say that a probability distribution $\rho \in \Delta_{\mathcal{C} \times \mathcal{V}}$ is $\epsilon$-*close* to $\rho^*$ if and only if every policy with statistical parity with respect to $\rho$ has $\epsilon$-approximate statistical parity with respect to $\rho^*$.

With these definitions in hand we have the following result:

**Theorem 5.3.** *There exists an instance of* SPEW *in which, with probability at least $1 - \delta$, there exists some $\rho \in \Delta_{\mathcal{C} \times \mathcal{V}}$, which is $\epsilon$-close to $\rho^*$, such that all the generated policies $\pi_t^*$ have statistical parity with respect to $\rho$ and, for any policy $\tilde{\pi}$ with statistical parity with respect to $\rho$, we have:*

$$\mathbb{E}\left[\sum_{t \in [T]} \ell_t(a_t)\right] \leq \sum_{t \in [T]} \sum_{a \in \mathcal{A}} \tilde{\pi}(c_t, v_t, a)\ell_t(a) + \tilde{\mathcal{O}}\left(\Psi(\tilde{\pi})\ln(1/\delta)(H/\epsilon)^2 + \left(\eta + \frac{H\Psi(\tilde{\pi})}{\eta}\right)\sqrt{KT}\right)$$

*If the determination of whether a context belongs to a given set of the hierarchical clustering takes $\mathcal{O}(1)$ time, and the hierarchical clustering needs not be explicitly maintained to determine this, then the per-trial time and space complexity of* SPEW *is $\mathcal{O}(KT)$.*

Note that unlike the previous examples, the time complexity is not dependent on the cardinality of the context set. The algorithm can be easily modified to obtain exact statistical parity (without the pairs $(c_t, v_t)$ necessarily being i.i.d.) when a distribution $\rho$ is known, as long as we can efficiently compute the probability masses of any set in the hierarchical clustering (for any given protected characteristic).

## 6 Batch Classification

Finally, we describe how SPEW can enforce statistical parity in the batch classification problem. Here there exists an unknown probability distribution $\hat{\rho} \in \Delta_{\mathcal{C} \times \mathcal{V} \times \mathcal{A}}$ and we are given a sequence:

$$\langle (c_t, v_t, b_t) \,|\, t \in [T] \rangle$$

of $T$ i.i.d. samples from $\hat{\rho}$. We are given a distribution $\rho \in \Delta_{\mathcal{C} \times \mathcal{V}}$ (which could be the empirical distribution of some unlabeled data). The aim is to construct a policy $\pi \in \mathcal{Q}$ which has statistical parity with respect to $\rho$ and has high *accuracy*:

$$\sum_{(c,v,a) \in \mathcal{C} \times \mathcal{V} \times \mathcal{A}} \hat{\rho}(c, v, a)\pi(c, v, a)$$

To solve this problem we first define, for all $t \in [T]$, the loss function $\ell_t$ by $\ell_t(a) := [\![a \neq b_t]\!]$ for all $a \in \mathcal{A}$ and then, with the appropriate instance of HEDGE, run SPEW on the sequence $\langle (c_t, v_t, \ell_t) \,|\, t \in [T] \rangle$ to produce a sequence of policies $\langle \pi_t^* \,|\, t \in [T] \rangle$ such that $\pi_t^*$ has statistical parity with respect to $\rho$ for all $t \in [T]$. Finally, we define the output policy:

$$\pi := \frac{1}{T}\sum_{t \in [T]} \pi_t^*$$

We note that via a HEDGE based doubling trick we can automatically tune the learning rate. Standard results on online-to-batch conversion [9] convert the regret bound given in Theorem 4.1 to a bound on the expected accuracy. Specifically, the difference between the expected accuracy of the algorithm and that of any given policy $\tilde{\pi}$ with statistical parity is equal to the bound on the regret divided by $T$. For trees this difference in expected accuracy is:

$$\tilde{\mathcal{O}}\left(\sqrt{(M + \Psi(\tilde{\pi}))\ln(N)K/T}\right)$$

where $\Psi$ is defined as in Section D.1.

## Acknowledgments

Research funded by the Defence Science and Technology Laboratory (Dstl) which is an executive agency of the UK Ministry of Defence providing world class expertise and delivering cutting-edge science and technology for the benefit of the nation and allies. The research supports the Autonomous Resilient Cyber Defence (ARCD) project within the Dstl Cyber Defence Enhancement programme.

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

## A   Literature Review

Algorithmic fairness has been a topic of much study. Notable pioneering works include [6], which introduced statistical parity, and [12] which introduced the notion of *individual fairness*. Similar to our batch result, [1] considered the problem of constructing a randomised binary classifier, with statistical parity, which is a convex combination of deterministic classifiers from some given hypothesis space. They solve the problem via a sequence of calls to a cost-sensitive classification oracle (for the particular hypothesis space) and give a generalisation error in terms of the Rademacher complexity of the hypthesis space. The work [11] considered finding the Bayes optimal classifier satisfying statistical parity approximately (with an arbitrary number of classes).

We now give a review of some of the research in the wide topic of achieving various forms of fairness in i.i.d. stochastic bandits. [32] considered the problem of eliminating societal bias when actions are partitioned into groups which are each subjected to societal bias. [19] considered, in the linear bandit setting, the problem of equalizing cumulative mean rewards across two protected groups of contexts. [31] considered the problem of, when actions are partitioned into groups, simulatenously achieving a certain exposure for each group and ensuring that, within each group, each action is

chosen with probability proportional to its *merit*. [21] considered the problem of ensuring that a better action (one with lower mean loss) never has a lower probability of being chosen than a worse action (one with higher mean loss), [33] considered the problem of choosing each action with a probability proportional to its merit, [22] considered the problem of treating similar actions similarly, and [28] considered the constraint of selecting each action at least a pre-specified number of times. A form of fairness with respect to instances was studied in [18] which considered the constraint of equalising the cumulative mean loss of all protected groups (all instances that have some given protected characteristic) in stochastic linear bandits. We note that, whilst this constraint will be desired in some situations, it is very different from statistical parity. Possibly the closest such work to ours is [4], which considered the i.i.d. stochastic online binary classification problem (classifying contexts as positive or negative) under the constraint that false negatives are equalised over two protected groups, under the assumption of the existence of an efficient oracle for empirical risk minimisation over a finite set of hypotheses. They specifically considered a partial feedback setting where the true class is only revealed when the positive class is chosen. However, the fairness constraints are only approximated and the algorithm is relatively inefficient: taking a per-trial time that is polynomial in the number of trials (the exponent is not given in the paper).

Other works on fairness in online learning include the work [5] which deals with combining expert advice with experts that are each fair with respect to some metric, so that the resulting policy is fair with respect to that metric. This is a very different problem to that studied in this paper: which utilises experts that do not satisfy statistical parity. Another notable work is [3] which considers the problem of enforcing individual fairness in online settings where no metric is given but instead there exists an auditor which identifies pairs of users treated unfairly.

Finally, as described in the introduction (where we discussed the works [8] and [34]), there has been much work on online learning subject to convex constraints (which are approximately satisfied in the long run). This line of research was pioneered by [23] and was extended to time-varying constraints in [24]. The work [7] considered both time-varying constraints and bandit feedback. The work [20] achieved bounds that can tradeoff between regret and constraint violation. The algorithm in [23] works by gradient descent/ascent on the Lagrangian, whilst our algorithm SPEW works in a fundamentally different way - by mapping a maintained policy (without statistical parity) to a policy with statistical parity and performing exponentiated gradient descent (on the maintained policy) with a surrogate for the resulting loss in terms of the original policy.

## B    Algorithm and Analysis

Here we describe and analyse the mechanics of SPEW. All lemmas stated in this section are proved in Section C. Since this section works entirely in the virtual context space we will refer to, in this section, virtual contexts and policies as *contexts* and *policies* respectively. We define $\mathcal{I} := \mathcal{C} \times \mathcal{X}$.

Let $\eta$, $\tilde{\pi}$, $\vartheta$ and $\vartheta^*$ be as in Theorem 4.1. Without loss of generality we can assume that:

$$\eta \leq \sqrt{T/K}$$

otherwise our regret bound would be vacuous. Recall that SPEW takes as input an algorithm (a.k.a. the *base algorithm*) for implementing Hedge with inductive bias $\vartheta$. For the base algorithm we will choose the learning rate:

$$\hat{\eta} := \frac{\eta}{\sqrt{KT}}$$

We will maintain $M$ instances of this algorithm, one for each protected characteristic $c \in \mathcal{C}$. For each $c \in \mathcal{C}$ let QUERY$[c]$ and UPDATE$[c]$ be the QUERY and UPDATE subroutines for the instance of the base algorithm associated with the protected characteristic $c$. Given some $c \in \mathcal{C}$, recall (from Section 4.1) that the instance of the base algorithm for protected characteristic $c$ implicitly maintains a distribution in $\Delta_{\mathcal{H}}$. Let $\vartheta_t(c, \circ)$ be the value of this distribution at the start of trial $t$, so that $\vartheta_1(c, \circ) = \vartheta$.

We now describe and analyse how SPEW behaves on trial $t$ and, in doing so, will bound the *instantaneous regret*:

$$r_t := \sum_{a \in \mathcal{A}} (\pi_t(c_t, x_t, a) - \tilde{\pi}(c_t, x_t, a))\ell_t(a)$$

## B.1 Fundamental Definitions

We now make some fundamental definitions. We first define:

$$\mathcal{S} := \mathbb{R}^{\mathcal{C} \times \mathcal{X} \times \mathcal{A}}$$

Given $(\phi, c, a) \in \mathcal{S} \times \mathcal{C} \times \mathcal{A}$ define:

$$\tilde{\omega}_t(\phi, c, a) := \sum_{x \in \mathcal{G}_t(c)} \mu_t(c, x)\phi(c, x, a) = \sum_{x \in \mathcal{X}} \mu_t(c, x)\phi(c, x, a)$$

and define:

$$\tilde{\delta}_t(\phi, c, a) := \max_{c' \in \mathcal{C}} \tilde{\omega}_t(\phi, c', a) - \tilde{\omega}_t(\phi, c, a)$$

Given $\phi \in \mathcal{S}$ define:

$$\tilde{\beta}_t(\phi) := \sum_{a \in \mathcal{A}} \max_{c \in \mathcal{C}} \tilde{\delta}_t(\phi, c, a)$$

Finally, we define $\mathcal{Z}_t$ to be the set of all $\phi \in \mathcal{S}$ in which for all $c, c' \in \mathcal{C}$ and $a \in \mathcal{A}$ we have:

$$\tilde{\omega}_t(\phi, c, a) = \tilde{\omega}_t(\phi, c', a)$$

We note that $\mathcal{F}(\mu_t) = \mathcal{P} \cap \mathcal{Z}_t$.

## B.2 Computing the Policy

Define the policy $\xi_t \in \mathcal{P}$ by:

$$\xi_t(c, x, a) := \sum_{e \in \mathcal{H}} [\![ e(x) = a ]\!] \vartheta_t(c, e)$$

for all $(c, x, a) \in \mathcal{C} \times \mathcal{X} \times \mathcal{A}$. We call $\xi_t$ the *raw* policy. For each $c \in \mathcal{C}$ SPEW needs only know $\xi_t(c, x, \circ)$ for $x \in \mathcal{G}_t(c)$, which it can compute by a call to QUERY$[c]$. We state the above equation as the following lemma:

**Lemma B.1.** *For all $(c, x, a) \in \mathcal{C} \times \mathcal{X} \times \mathcal{A}$ we have:*

$$\xi_t(c, x, a) = \sum_{e \in \mathcal{H}} [\![ e(x) = a ]\!] \vartheta_t(c, e)$$

We now describe the conversion of our raw policy $\xi_t$ into our fair policy $\pi_t$. We call this conversion process *policy processing*. First define, for all $a \in \mathcal{A}$ and $c \in \mathcal{C}$, the quantities:

$$\delta_t(c, a) := \tilde{\delta}_t(\xi_t, c, a) \quad ; \quad \beta_t := \tilde{\beta}_t(\xi_t)$$

We define a function $\psi_t \in \mathcal{S}$ such that for any $(c, x, a) \in \mathcal{C} \times \mathcal{X} \times \mathcal{A}$ we have:

$$\psi_t(c, x, a) := \frac{\xi_t(c, x, a) + \delta_t(c, a)}{1 + \beta_t}$$

The function $\psi_t$ has the following properties:

**Lemma B.2.** *We have that $\psi_t \in \mathcal{Z}_t$. In addition we have, for all $(c, x) \in \mathcal{I}$, that:*

$$\sum_{a \in \mathcal{A}} \psi_t(c, x, a) \le 1$$

*and that $\psi_t(c, x, a) \ge 0$ for all $a \in \mathcal{A}$.*

The above lemma shows that the function $\psi_t$ satisfies most of our constraints. However, it is not necessarily in $\mathcal{P}$ as the functions $\psi_t(c, x, \circ)$ may not lie on the simplex $\Delta_{\mathcal{A}}$. In order to convert it into $\pi_t$ we simply add, to each of the functions $\psi_t(c, x, \circ)$, some the uniform distribution. Specifically, for all $(c, x, a) \in \mathcal{C} \times \mathcal{X} \times \mathcal{A}$ we define:

$$\pi_t(c, x, a) := \psi_t(c, x, a) + \frac{1}{K}\left(1 - \sum_{a' \in \mathcal{A}} \psi_t(c, x, a')\right)$$

The following lemma, which follows from Lemma B.2, confirms that $\pi_t$ does indeed lie in $\mathcal{F}(\mu_t)$.

**Lemma B.3.** *We have:*
$$\pi_t \in \mathcal{F}(\mu_t)$$

We now give two additional lemmas that are crucial to our analysis. The first lemma follows from Lemma B.2.

**Lemma B.4.** *For all $(c, x, a) \in \mathcal{C} \times \mathcal{X} \times \mathcal{A}$ we have:*
$$\pi_t(c, x, a) \geq \frac{\xi_t(c, x, a)}{1 + \beta_t}$$

The next lemma follows from lemmas B.3 and B.4

**Lemma B.5.** *For all $(c, x) \in \mathcal{I}$ we have:*
$$\sum_{a \in \mathcal{A}} \max\{0, \pi_t(c, x, a) - \xi_t(c, x, a)\} \leq \beta_t$$

We have now derived all the properties of $\pi_t$ needed to proceed with the algorithm and analysis.

### B.3 The Objective Function and its Pseudo-Gradient

Now that we have shown how to compute $\pi_t$ we turn to the update of the base algorithm instances at the end of the trial. In order to do this we need a convex objective function $y_t : \mathcal{S} \to \mathbb{R}$ which we note will never actually be known by Learner. Our objective function is defined such that for all $\phi \in \mathcal{S}$ we have:
$$y_t(\phi) := [\![\beta_t \leq 1]\!] \sum_{a \in \mathcal{A}} \ell_t(a)\phi(c_t, x_t, a) + \tilde{\beta}_t(\phi)$$

The reason we have chosen such an objective function is due to the following two lemmas, where Lemma B.6 follows from Lemma B.5. We note that the appearance of $[\![\beta_t \leq 1]\!]$ in the definition of $y_t$ is to ensure another crucial lemma, which we shall present later.

**Lemma B.6.** *We have:*
$$y_t(\xi_t) \geq \sum_{a \in \mathcal{A}} \pi_t(c_t, x_t, a)\ell_t(a)$$

**Lemma B.7.** *We have:*
$$y_t(\tilde{\pi}) \leq \sum_{a \in \mathcal{A}} \tilde{\pi}(c_t, x_t, a)\ell_t(a)$$

We will show that $y_t$ is indeed convex by showing that it has a sub-gradient at every point in $\mathcal{S}$. In order to give such a sub-gradient we first define, for all $(\phi, a) \in \mathcal{S} \times \mathcal{A}$, the quantities:
$$\tilde{\kappa}_t(\phi, a) := \operatorname{argmax}_{c \in \mathcal{C}} \tilde{\omega}_t(\phi, c, a) \quad ; \quad \tilde{\kappa}'_t(\phi, a) := \operatorname{argmin}_{c \in \mathcal{C}} \tilde{\omega}_t(\phi, c, a)$$

where ties are broken arbitrarily. We then define $g_t : \mathcal{S} \times \mathcal{C} \times \mathcal{X} \times \mathcal{A} \to \mathbb{R}$ such that for all $(\phi, c, x, a) \in \mathcal{S} \times \mathcal{C} \times \mathcal{X} \times \mathcal{A}$ we have:
$$g_t(\phi, c, x, a) := [\![\beta_t \leq 1]\!][\![(c, x) = (c_t, x_t)]\!]\ell_t(a) + [\![c = \tilde{\kappa}_t(\phi, a)]\!]\mu_t(c, x) - [\![c = \tilde{\kappa}'_t(\phi, a)]\!]\mu_t(c, x)$$

The next lemma shows that for all $\phi \in \mathcal{S}$ we have that $g_t(\phi, \circ, \circ, \circ)$ is a sub-gradient of $y_t$ at $\phi$.

**Lemma B.8.** *For any $\phi, \phi' \in \mathcal{S}$ we have:*
$$y_t(\phi) - y_t(\phi') \leq \sum_{c \in \mathcal{C}} \sum_{x \in \mathcal{X}} \sum_{a \in \mathcal{A}} g_t(\phi, c, x, a)(\phi(c, x, a) - \phi'(c, x, a))$$

Due to the fact that we don't know the entire function $\ell_t$, we can't compute the sub-gradient $g_t(\xi_t, \circ, \circ, \circ)$ so instead, will borrow the technique, from EXP4, of using $[\![a = a_t]\!]\ell_t(a_t)/\pi_t(c_t, x_t, a_t)$ as an unbiased estimator of $\ell_t(a)$. This gives us the following function $\lambda_t$ that we call the *pseudo-gradient*. $\lambda_t \in \mathcal{S}$ is defined such that for all $(c, x, a) \in \mathcal{C} \times \mathcal{X} \times \mathcal{A}$ we have:
$$\lambda_t(c, x, a) := [\![\beta_t \leq 1]\!][\![(c, x, a) = (c_t, x_t, a_t)]\!]\frac{\ell_t(a_t)}{\pi_t(c_t, x_t, a_t)}$$
$$+ [\![c = \tilde{\kappa}_t(\xi_t, a)]\!]\mu_t(c, x) - [\![c = \tilde{\kappa}'_t(\xi_t, a)]\!]\mu_t(c, x)$$

The following lemma states that $\lambda_t$ is an unbiased estimator of the sub-gradient $g_t(\xi_t, \circ, \circ, \circ)$.

**Lemma B.9.** *For all $(c, x, a) \in \mathcal{C} \times \mathcal{X} \times \mathcal{A}$ we have:*

$$\mathbb{E}[\lambda_t(c, x, a) \,|\, \xi_t] = g_t(\xi_t, c, x, a)$$

Note that for the full-information online and batch classification versions of SPEW we can use the true sub-gradient instead of $\lambda_t$ in order to give a deterministic algorithm.

Lemmas B.6, B.7, B.8 and B.9 combine to give us the following lemma.

**Lemma B.10.** *We have:*

$$\mathbb{E}\left[\sum_{c \in \mathcal{C}} \sum_{x \in \mathcal{X}} \sum_{a \in \mathcal{A}} (\xi_t(c, x, a) - \tilde{\pi}(c, x, a))\lambda_t(c, x, a) \,\middle|\, \xi_t\right] \geq r_t$$

Now define $\nu_t : \mathcal{C} \times \mathcal{H} \to \mathbb{R}$ such that for all $(c, e) \in \mathcal{C} \times \mathcal{H}$ we have:

$$\nu_t(c, e) := \sum_{x \in \mathcal{X}} \lambda_t(c, x, e(x))$$

Lemmas B.1 and B.10 combine to give us the following lemma.

**Lemma B.11.** *We have:*

$$\mathbb{E}\left[\sum_{c \in \mathcal{C}} \sum_{e \in \mathcal{H}} (\vartheta_t(c, e) - \vartheta^*(c, e))\nu_t(c, e) \,\middle|\, \vartheta_t\right] \geq r_t$$

The reason that the term $[\![\beta_t \leq 1]\!]$ appears in the definition of our objective function $y_t$ is to ensure the following crucial property of the pseudo-gradient, which follows from Lemma B.4.

**Lemma B.12.** *We have:*

$$\mathbb{E}\left[\sum_{c \in \mathcal{C}} \sum_{e \in \mathcal{H}} \vartheta_t(c, e)\nu_t(c, e)^2 \,\middle|\, \vartheta_t\right] \leq 8K$$

Finally we have the following lemma:

**Lemma B.13.** *For all $(c, e) \in \mathcal{C} \times \mathcal{H}$ we have:*

$$\nu_t(c, e) \geq -K$$

We have now derived all the properties of $\nu_t$ needed to progress to the next stage of the algorithm and analysis.

## B.4 Hedge Update

To update we run, for each $c \in \mathcal{C}$ and $x \in \mathcal{G}_t(c)$, the function:

$$\text{UPDATE}[c](x, \lambda_t(c, x, \circ))$$

We have the following lemma.

**Lemma B.14.** *For all $(c, e) \in \mathcal{C} \times \mathcal{H}$ we have:*

$$\vartheta_{t+1}(c, e) = \frac{\vartheta_t(c, e) \exp(-\hat{\eta}\nu_t(c, e))}{\sum_{e' \in \mathcal{H}} \vartheta_t(c, e') \exp(-\hat{\eta}\nu_t(c, e'))}$$

We define the *relative entropy* $B : \Delta_{\mathcal{H}} \times \Delta_{\mathcal{H}} \to \mathbb{R}$ such that for all $\beta, \beta' \in \Delta_{\mathcal{H}}$ we have:

$$B(\beta, \beta') := \sum_{e \in \mathcal{H}} \beta(e) \ln\left(\frac{\beta(e)}{\beta'(e)}\right)$$

Lemma B.13 and Lemma B.14 give us the following lemma.

**Lemma B.15.** *For all $c \in \mathcal{C}$ we have:*

$$B(\vartheta^*, \vartheta_t(c, \circ)) - B(\vartheta^*, \vartheta_{t+1}(c, \circ)) \geq \hat{\eta} \sum_{e \in \mathcal{H}} (\vartheta_t(c, e) - \vartheta^*(c, e))\nu_t(c, e) - \hat{\eta}^2 \sum_{e \in \mathcal{H}} \vartheta_t(c, e)\nu_t(c, e)^2$$

This lemma combines with lemmas B.11 and B.12 to give the following lemma.

**Lemma B.16.** *We have:*

$$\mathbb{E}\left[\sum_{c \in \mathcal{C}} (B(\vartheta^*, \vartheta_t(c, \circ)) - B(\vartheta^*, \vartheta_{t+1}(c, \circ))) \,\middle|\, \vartheta_t\right] \geq \hat{\eta} r_t - 8\hat{\eta}^2 K$$

This completes the description and analysis of trial $t$.

## B.5   Regret Bound

To get the overall regret bound we take expectations on Lemma B.16 and sum over all $t \in [T]$, which gives us the following lemma.

**Lemma B.17.** *We have:*

$$\sum_{t \in [T]} \mathbb{E}[r_t] \leq (8\eta - \Phi/\eta)\sqrt{K/T}$$

*where:*

$$\Phi := \sum_{c \in \mathcal{C}} \sum_{e \in \mathcal{H}} \vartheta^*(c, e) \ln\left(\frac{\vartheta^*(c, e)}{\vartheta(e)}\right)$$

# C   Proofs of Lemmas

We now prove, in order, all of the lemmas given in this Appendix B.

## C.1   Lemma B.1

This is immediately clear from the construction of $\xi_t$ and the description of HEDGE given in Section 4.1.

## C.2   Lemma B.2

We first show that $\psi_t \in \mathcal{Z}_t$. To show this consider some $c \in \mathcal{C}$ and $a \in \mathcal{A}$. We have:

$$\begin{aligned}
(1 + \beta_t)\tilde{\omega}_t(\psi_t, c, a) &= (1 + \beta_t) \sum_{x \in \mathcal{X}} \mu_t(c, x)\psi_t(c, x, a) \\
&= \sum_{x \in \mathcal{X}} \mu_t(c, x)(\xi_t(c, x, a) + \delta_t(c, a)) \\
&= \sum_{x \in \mathcal{X}} \mu_t(c, x)\xi_t(c, x, a) + \delta_t(c, a) \sum_{x \in \mathcal{X}} \mu_t(c, x) \\
&= \tilde{\omega}_t(\xi_t, c, a) + \delta_t(c, a) \\
&= \tilde{\omega}_t(\xi_t, c, a) + \tilde{\delta}_t(\xi_t, c, a) \\
&= \tilde{\omega}_t(\xi_t, c, a) + \max_{c' \in \mathcal{C}} \tilde{\omega}_t(\xi_t, c', a) - \tilde{\omega}_t(\xi_t, c, a) \\
&= \max_{c' \in \mathcal{C}} \tilde{\omega}_t(\xi_t, c', a)
\end{aligned}$$

Since this is independent of $c$ we have that $\psi_t \in \mathcal{Z}_t$ as required.

We now show that, given $(c, x) \in \mathcal{I}$, we have $\sum_{a \in \mathcal{A}} \psi_t(c, x, a) \leq 1$. This is true since $\xi_t \in \mathcal{P}$ and hence:

$$
\begin{aligned}
(1 + \beta_t) \sum_{a \in \mathcal{A}} \psi_t(c, x, a) &= \sum_{a \in \mathcal{A}} \xi_t(c, x, a) + \sum_{a \in \mathcal{A}} \delta_t(c, a) \\
&= 1 + \sum_{a \in \mathcal{A}} \tilde{\delta}_t(\xi_t, c, a) \\
&\leq 1 + \sum_{a \in \mathcal{A}} \max_{c' \in \mathcal{C}} \tilde{\delta}_t(\xi_t, c', a) \\
&= 1 + \tilde{\beta}_t(\xi_t) \\
&\leq (1 + \beta_t)
\end{aligned}
$$

as required.

The fact that $\psi_t(c, x, a) \geq 0$ for all $(c, x, a) \in \mathcal{C} \times \mathcal{X} \times \mathcal{A}$ follows directly from the fact that, since $\xi_t \in \mathcal{P}$, we have that $\delta_t(c, a)$ and $\beta_t$ and $\xi_t(c, x, a)$ are all non-negative.

### C.3 Lemma B.3

We first show that $\pi_t \in \mathcal{P}$. To show this consider any $(c, x) \in \mathcal{I}$. By Lemma B.2 we have that $\sum_{a \in \mathcal{A}} \psi_t(c, x, a) \leq 1$ and hence:

$$
1 - \sum_{a' \in \mathcal{A}} \psi_t(c, x, a') \geq 0
$$

For all $a \in \mathcal{A}$ we have, by Lemma B.2, that $\psi_t(c, x, a) \geq 0$ and hence the above equation implies that:

$$
\pi_t(c, x, a) = \psi_t(c, x, a) + \frac{1}{K} \left( 1 - \sum_{a' \in \mathcal{A}} \psi_t(c, x, a') \right) \geq 0 + 0 = 0
$$

Finally note that:

$$
\begin{aligned}
\sum_{a \in \mathcal{A}} \pi_t(c, x, a) &= \sum_{a \in \mathcal{A}} \psi_t(c, x, a) + \frac{1}{K} \sum_{a \in \mathcal{A}} \left( 1 - \sum_{a' \in \mathcal{A}} \psi_t(c, x, a') \right) \\
&= \sum_{a \in \mathcal{A}} \psi_t(c, x, a) + \left( 1 - \sum_{a' \in \mathcal{A}} \psi_t(c, x, a') \right) \\
&= 1
\end{aligned}
$$

Since these equations hold for all $(c, x) \in \mathcal{I}$, we have shown that $\pi_t \in \mathcal{P}$. Hence, all we now need to show is that $\pi_t \in \mathcal{Z}_t$. To show this, note that by Lemma B.2 we have that $\psi_t \in \mathcal{Z}_t$. Hence, there exists a function $z : \mathcal{A} \rightarrow \mathbb{R}$ such that for all $(c, a) \in \mathcal{C} \times \mathcal{A}$ we have:

$$
\sum_{x \in \mathcal{X}} \mu_t(c, x) \psi_t(c, x, a) = \tilde{\omega}_t(\psi_t, c, a) = z(a)
$$

So for all $(c, a) \in \mathcal{C} \times \mathcal{A}$ we have:

$$
\begin{aligned}
\tilde{\omega}_t(\pi_t, c, a) &= \sum_{x \in \mathcal{X}} \mu_t(c, x) \pi_t(c, x, a) \\
&= \sum_{x \in \mathcal{X}} \mu_t(c, x) \psi_t(c, x, a) + \sum_{x \in \mathcal{X}} \frac{\mu_t(c, x)}{K} \left( 1 - \sum_{a' \in \mathcal{A}} \psi_t(c, x, a') \right) \\
&= z(a) + \frac{1}{K} \sum_{x \in \mathcal{X}} \mu_t(c, x) - \frac{1}{K} \sum_{a' \in \mathcal{A}} \sum_{x \in \mathcal{X}} \mu_t(c, x) \psi_t(c, x, a') \\
&= z(a) + \frac{1}{K} - \frac{1}{K} \sum_{a' \in \mathcal{A}} z(a')
\end{aligned}
$$

Since this is independent of $c$ we have $\pi_t \in \mathcal{Z}_t$ as required. This completes the proof.

## C.4 Lemma B.4

Choose any $(c, x, a) \in \mathcal{C} \times \mathcal{X} \times \mathcal{A}$. By Lemma B.2 we have $\sum_{a \in \mathcal{A}} \psi_t(c, x, a) \leq 1$ so since $\delta_t(c, a)$ and $\beta_t$ and $\xi_t(c, x, a)$ are all non-negative, we have that:

$$
\begin{aligned}
\pi_t(c, x, a) &= \psi_t(c, x, a) + \frac{1}{K}\left(1 - \sum_{a' \in \mathcal{A}} \psi_t(c, x, a')\right) \\
&\geq \psi_t(c, x, a) + \frac{1}{K}(1 - 1) \\
&= \psi_t(c, x, a) \\
&= \frac{\xi_t(c, x, a) + \delta_t(c, a)}{1 + \beta_t} \\
&\geq \frac{\xi_t(c, x, a)}{1 + \beta_t}
\end{aligned}
$$

as required.

## C.5 Lemma B.5

Take any $(c, x, a) \in \mathcal{C} \times \mathcal{X} \times \mathcal{A}$. By Lemma B.4 we have:

$$
0 \leq \pi_t(c, x, a) - \frac{\xi_t(c, x, a)}{1 + \beta_t}
$$

Since $\beta_t$ and $\xi_t(c, x, a)$ are non-negative we also have:

$$
\pi_t(c, x, a) - \xi_t(c, x, a) \leq \pi_t(c, x, a) - \frac{\xi_t(c, x, a)}{1 + \beta_t}
$$

and hence we have shown that:

$$
\max\{0, \pi_t(c, x, a) - \xi_t(c, x, a)\} \leq \pi_t(c, x, a) - \frac{\xi_t(c, x, a)}{1 + \beta_t}
$$

Now take any $(c, x) \in \mathcal{I}$. By lemmas B.1 and B.3 we have that both $\pi_t$ and $\xi_t$ are in $\mathcal{P}$. Hence, by the above equation we have, since $\beta_t \geq 0$, that:

$$
\begin{aligned}
\sum_{a \in \mathcal{A}} \max\{0, \pi_t(c, x, a) - \xi_t(c, x, a)\} &\leq \sum_{a \in \mathcal{A}} \left(\pi_t(c, x, a) - \frac{\xi_t(c, x, a)}{1 + \beta_t}\right) & (1) \\
&= \sum_{a \in \mathcal{A}} \pi_t(c, x, a) - \frac{1}{1 + \beta_t} \sum_{a \in \mathcal{A}} \xi_t(c, x, a) & (2) \\
&= 1 - \frac{1}{1 + \beta_t} & (3) \\
&= \frac{\beta_t}{1 + \beta_t} & (4) \\
&\leq \beta_t & (5)
\end{aligned}
$$

as required.

## C.6 Lemma B.6

We consider two cases. First consider the case that $\beta_t > 1$. In this case we have, since $\xi_t \in \mathcal{P}$ and $\ell_t(a) \leq 1$ for all $a \in \mathcal{A}$, that:

$$
\begin{aligned}
y_t(\xi_t) = [\![\beta_t \leq 1]\!] \sum_{a \in \mathcal{A}} \ell_t(a)\xi_t(c_t, x_t, a) + \tilde{\beta}_t(\xi_t) \\
= \tilde{\beta}_t(\xi_t) \\
= \beta_t \\
> 1 \\
= \sum_{a \in \mathcal{A}} \pi_t(c_t, x_t, a) \\
\geq \sum_{a \in \mathcal{A}} \pi_t(c_t, x_t, a)\ell_t(a)
\end{aligned}
$$

Now consider the case that $\beta_t \leq 1$. Since $\ell_t(a) \in [0, 1]$ for all $a \in \mathcal{A}$, we have, by Lemma B.5, that:

$$
\begin{aligned}
\sum_{a \in \mathcal{A}} \ell_t(a)\pi_t(c_t, x_t, a) - \sum_{a \in \mathcal{A}} \ell_t(a)\xi_t(c_t, x_t, a) = \sum_{a \in \mathcal{A}} \ell_t(a)(\pi_t(c_t, x_t, a) - \xi_t(c_t, x_t, a)) \\
\leq \sum_{a \in \mathcal{A}} \ell_t(a) \max\{0, \pi_t(c_t, x_t, a) - \xi_t(c_t, x_t, a)\} \\
\leq \sum_{a \in \mathcal{A}} \max\{0, \pi_t(c_t, x_t, a) - \xi_t(c_t, x_t, a)\} \\
\leq \beta_t
\end{aligned}
$$

Hence, we have that:

$$
\begin{aligned}
y_t(\xi_t) = [\![\beta_t \leq 1]\!] \sum_{a \in \mathcal{A}} \ell_t(a)\xi_t(c_t, x_t, a) + \tilde{\beta}_t(\xi_t) \\
= \sum_{a \in \mathcal{A}} \ell_t(a)\xi_t(c_t, x_t, a) + \beta_t \\
\geq \sum_{a \in \mathcal{A}} \ell_t(a)\xi_t(c_t, x_t, a) + \left( \sum_{a \in \mathcal{A}} \ell_t(a)\pi_t(c_t, x_t, a) - \sum_{a \in \mathcal{A}} \ell_t(a)\xi_t(c_t, x_t, a) \right) \\
= \sum_{a \in \mathcal{A}} \ell_t(a)\pi_t(c_t, x_t, a)
\end{aligned}
$$

So in either case we have the result.

## C.7 Lemma B.7

Since $\tilde{\pi} \in \mathcal{F}(\mu_t)$ we have $\tilde{\pi}_t \in \mathcal{Z}_t$ which implies that there exists a function $z : \mathcal{A} \to \mathbb{R}$ such that for all $(c, a) \in \mathcal{C} \times \mathcal{A}$ we have $\tilde{\omega}_t(\tilde{\pi}, c, a) = z(a)$. Hence, we have, for all $(c, a) \in \mathcal{C} \times \mathcal{A}$, that:

$$
\tilde{\delta}_t(\tilde{\pi}, c, a) = \max_{c' \in \mathcal{C}} \tilde{\omega}_t(\tilde{\pi}, c', a) - \tilde{\omega}_t(\tilde{\pi}, c, a) = z(a) - z(a) = 0
$$

Since $\ell_t(a)$ and $\tilde{\pi}(c_t, x_t, a)$ are non-negative for all $a \in \mathcal{A}$, this implies that:

$$
\begin{aligned}
y_t(\tilde{\pi}) := [\![\beta_t \leq 1]\!] \sum_{a \in \mathcal{A}} \ell_t(a)\tilde{\pi}(c_t, x_t, a) + \tilde{\beta}_t(\tilde{\pi}) \\
\leq \sum_{a \in \mathcal{A}} \ell_t(a)\tilde{\pi}(c_t, x_t, a) + \sum_{a \in \mathcal{A}} \max_{c \in \mathcal{C}} \tilde{\delta}_t(\tilde{\pi}, c, a) \\
= \sum_{a \in \mathcal{A}} \ell_t(a)\tilde{\pi}(c_t, x_t, a)
\end{aligned}
$$

as required.

## C.8 Lemma B.8

Take any $a \in \mathcal{A}$. Given $\phi^* \in \mathcal{S}$ and $c' \in \mathcal{C}$ we have:

$$\sum_{(c,x) \in \mathcal{I}} [\![c = c']\!] \mu_t(c,x) \phi^*(c,x,a) = \sum_{x \in \mathcal{X}} \mu_t(c',x) \phi^*(c',x,a) = \tilde{\omega}_t(\phi^*, c', a)$$

which implies that:

$$\sum_{(c,x) \in \mathcal{I}} [\![c = \tilde{\kappa}_t(\phi,a)]\!] \mu_t(c,x) \phi(c,x,a) = \max_{c' \in \mathcal{C}} \tilde{\omega}_t(\phi, c', a) \tag{6}$$

$$\sum_{(c,x) \in \mathcal{I}} [\![c = \tilde{\kappa}'_t(\phi,a)]\!] \mu_t(c,x) \phi(c,x,a) = \min_{c' \in \mathcal{C}} \tilde{\omega}_t(\phi, c', a) \tag{7}$$

$$\sum_{(c,x) \in \mathcal{I}} [\![c = \tilde{\kappa}_t(\phi,a)]\!] \mu_t(c,x) \phi'(c,x,a) \leq \max_{c' \in \mathcal{C}} \tilde{\omega}_t(\phi', c', a) \tag{8}$$

$$\sum_{(c,x) \in \mathcal{I}} [\![c = \tilde{\kappa}'_t(\phi,a)]\!] \mu_t(c,x) \phi'(c,x,a) \geq \min_{c' \in \mathcal{C}} \tilde{\omega}_t(\phi', c', a) \tag{9}$$

For all $a \in \mathcal{A}$, equations (6) and (7) imply that:

$$\sum_{(c,x) \in \mathcal{I}} g_t(\phi, c, x, a) \phi(c, x, a)$$

$$= \sum_{(c,x) \in \mathcal{I}} [\![\beta_t \leq 1]\!] [\![(c,x) = (c_t, x_t)]\!] \ell_t(a) \phi(c,x,a) + \max_{c' \in \mathcal{C}} \tilde{\omega}_t(\phi, c', a) - \min_{c' \in \mathcal{C}} \tilde{\omega}_t(\phi, c', a)$$

$$= \sum_{(c,x) \in \mathcal{I}} [\![\beta_t \leq 1]\!] [\![(c,x) = (c_t, x_t)]\!] \ell_t(a) \phi(c,x,a) + \max_{c^\dagger \in \mathcal{C}} \left( \max_{c' \in \mathcal{C}} \tilde{\omega}_t(\phi, c', a) - \tilde{\omega}_t(\phi, c^\dagger, a) \right)$$

$$= [\![\beta_t \leq 1]\!] \ell_t(a) \phi(c_t, x_t, a) + \max_{c^\dagger \in \mathcal{C}} \tilde{\delta}_t(\phi, c^\dagger a)$$

This means that:

$$\sum_{a \in \mathcal{A}} \sum_{(c,x) \in \mathcal{I}} g_t(\phi, c, x, a) \phi(c, x, a) = \sum_{a \in \mathcal{A}} [\![\beta_t \leq 1]\!] \ell_t(a) \phi(c_t, x_t, a) + \sum_{a \in \mathcal{A}} \max_{c^\dagger \in \mathcal{C}} \tilde{\delta}_t(\phi, c^\dagger, a)$$

$$= [\![\beta_t \leq 1]\!] \sum_{a \in \mathcal{A}} \ell_t(a) \phi(c_t, x_t, a) + \tilde{\beta}_t(\phi)$$

$$= y_t(\phi) \tag{10}$$

Similarly, for all $a \in \mathcal{A}$, equations (8) and (9) imply that:

$$\sum_{(c,x) \in \mathcal{I}} g_t(\phi, c, x, a) \phi'(c, x, a)$$

$$\leq \sum_{(c,x) \in \mathcal{I}} [\![\beta_t \leq 1]\!] [\![(c,x) = (c_t, x_t)]\!] \ell_t(a) \phi'(c,x,a) + \max_{c' \in \mathcal{C}} \tilde{\omega}_t(\phi', c', a) - \min_{c' \in \mathcal{C}} \tilde{\omega}_t(\phi', c', a)$$

$$= \sum_{(c,x) \in \mathcal{I}} [\![\beta_t \leq 1]\!] [\![(c,x) = (c_t, x_t)]\!] \ell_t(a) \phi'(c,x,a) + \max_{c^\dagger \in \mathcal{C}} \left( \max_{c' \in \mathcal{C}} \tilde{\omega}_t(\phi', c', a) - \tilde{\omega}_t(\phi', c^\dagger, a) \right)$$

$$= [\![\beta_t \leq 1]\!] \ell_t(a) \phi'(c_t, x_t, a) + \max_{c^\dagger \in \mathcal{C}} \tilde{\delta}_t(\phi, c^\dagger, a)$$

This means that:

$$\sum_{a \in \mathcal{A}} \sum_{(c,x) \in \mathcal{I}} g_t(\phi, c, x, a) \phi'(c, x, a) \leq \sum_{a \in \mathcal{A}} [\![\beta_t \leq 1]\!] \ell_t(a) \phi'(c_t, x_t, a) + \sum_{a \in \mathcal{A}} \max_{c^\dagger \in \mathcal{C}} \tilde{\delta}_t(\phi, c^\dagger, a)$$

$$= [\![\beta_t \leq 1]\!] \sum_{a \in \mathcal{A}} \ell_t(a) \phi'(c_t, x_t, a) + \tilde{\beta}_t(\phi')$$

$$= y_t(\phi') \tag{11}$$

Equations 10 and 11 give us:

$$\sum_{(c,x)\in\mathcal{I}}\sum_{a\in\mathcal{A}} g_t(\phi,c,x,a)(\phi(c,x,a)-\phi'(c,x,a)) = \sum_{(c,x)\in\mathcal{I}}\sum_{a\in\mathcal{A}} g_t(\phi,c,x,a)\phi(c,x,a) - \sum_{(c,x)\in\mathcal{I}}\sum_{a\in\mathcal{A}} g_t(\phi,c,x,a)\phi'(c,x,a)$$

$$\geq y_t(\phi) - y_t(\phi')$$

as required.

### C.9 Lemma B.9

In this proof we use, throughout, the property of linearity of expectation. Take any $a \in \mathcal{A}$. Since $x_t$, $c_t$, $\pi_t$ and $\ell_t(a)$ are deterministic when conditioned on $\xi_t$ we have:

$$\mathbb{E}\left[\llbracket a = a_t\rrbracket \frac{\ell_t(a_t)}{\pi_t(c_t,x_t,a_t)}\,\middle|\,\xi_t\right] = \mathbb{E}\left[\llbracket a = a_t\rrbracket \frac{\ell_t(a)}{\pi_t(c_t,x_t,a)}\,\middle|\,\xi_t\right]$$

$$= \frac{\ell_t(a)}{\pi_t(c_t,x_t,a)}\mathbb{E}[\llbracket a = a_t\rrbracket\,|\,\xi_t]$$

$$= \frac{\ell_t(a)}{\pi_t(c_t,x_t,a)}\mathbb{P}[a = a_t\,|\,\xi_t]$$

$$= \frac{\ell_t(a)}{\pi_t(c_t,x_t,a)}\pi_t(c_t,x_t,a)$$

$$= \ell_t(a) \tag{12}$$

Which immediately gives us the result.

### C.10 Lemma B.10

By lemmas B.6 and B.7 we have:

$$r_t = \sum_{a\in\mathcal{A}}\pi_t(c_t,x_t,a)\ell_t(a) - \sum_{a\in\mathcal{A}}\tilde{\pi}(c_t,x_t,a)\ell_t(a) \leq y_t(\xi_t) - y_t(\tilde{\pi}) \tag{13}$$

and by lemmas B.8 and B.9 and the linearity of expectation we have, since $\tilde{\pi}$ is independent of $\xi_t$, that:

$$y_t(\xi_t) - y_t(\tilde{\pi}) \leq \sum_{(c,x)\in\mathcal{I}}\sum_{a\in\mathcal{A}} g_t(\xi_t,c,x,a)(\xi_t(c,x,a)-\tilde{\pi}(c,x,a))$$

$$= \sum_{(c,x)\in\mathcal{I}}\sum_{a\in\mathcal{A}} \mathbb{E}[\lambda_t(c,x,a)\,|\,\xi_t](\xi_t(c,x,a)-\tilde{\pi}(c,x,a))$$

$$= \mathbb{E}\left[\sum_{(c,x)\in\mathcal{I}}\sum_{a\in\mathcal{A}}(\xi_t(c,x,a)-\tilde{\pi}(c,x,a))\lambda_t(c,x,a)\,\middle|\,\xi_t\right] \tag{14}$$

Combining equations (13) and (14) gives us the result.

### C.11 Lemma B.11

By Lemma B.1 we have:

$$\sum_{c\in\mathcal{C}}\sum_{e\in\mathcal{H}}\vartheta_t(c,e)\nu_t(c,e) = \sum_{c\in\mathcal{C}}\sum_{e\in\mathcal{H}}\vartheta_t(c,e)\sum_{x\in\mathcal{X}}\lambda_t(c,x,e(x))$$

$$= \sum_{c\in\mathcal{C}}\sum_{e\in\mathcal{H}}\vartheta_t(c,e)\sum_{x\in\mathcal{X}}\sum_{a\in\mathcal{A}}\llbracket e(x)=a\rrbracket\lambda_t(c,x,a)$$

$$= \sum_{(c,x)\in\mathcal{I}}\sum_{a\in\mathcal{A}}\lambda_t(c,x,a)\sum_{e\in\mathcal{H}}\vartheta_t(c,e)\llbracket e(x)=a\rrbracket$$

$$= \sum_{(c,x)\in\mathcal{I}}\sum_{a\in\mathcal{A}}\lambda_t(c,x,a)\xi_t(c,x,a)$$

and by definition of $\vartheta^*$ we have:

$$
\begin{aligned}
\sum_{c\in\mathcal{C}}\sum_{e\in\mathcal{H}}\vartheta^*(c,e)\nu_t(c,e) &= \sum_{c\in\mathcal{C}}\sum_{e\in\mathcal{H}}\vartheta^*(c,e)\sum_{x\in\mathcal{X}}\lambda_t(c,x,e(x)) \\
&= \sum_{c\in\mathcal{C}}\sum_{e\in\mathcal{H}}\vartheta^*(c,e)\sum_{x\in\mathcal{X}}\sum_{a\in\mathcal{A}}[\![e(x)=a]\!]\lambda_t(c,x,a) \\
&= \sum_{(c,x)\in\mathcal{I}}\sum_{a\in\mathcal{A}}\lambda_t(c,x,a)\sum_{e\in\mathcal{H}}\vartheta^*(c,e)[\![e(x)=a]\!] \\
&= \sum_{(c,x)\in\mathcal{I}}\sum_{a\in\mathcal{A}}\lambda_t(c,x,a)\tilde{\pi}(c,x,a)
\end{aligned}
$$

so since $\xi_t$ is derived solely from $\vartheta_t$ (and deterministic objects) we have:

$$
\mathbb{E}\left[\sum_{c\in\mathcal{C}}\sum_{e\in\mathcal{H}}(\vartheta_t(c,e)-\vartheta^*(c,e))\nu_t(c,e)\,\middle|\,\vartheta_t\right] = \mathbb{E}\left[\sum_{c\in\mathcal{C}}\sum_{x\in\mathcal{X}}\sum_{a\in\mathcal{A}}(\xi_t(c,x,a)-\tilde{\pi}(c,x,a)\lambda_t(c,x,a)\,\middle|\,\xi_t\right]
$$

so Lemma B.10 gives us the result.

### C.12 Lemma B.12

Since $(z+\hat{z})^2 \le 2z^2 + 2\hat{z}^2$ for all $z,\hat{z}\in\mathbb{R}$ and since $(z-\hat{z})^2 \le z^2 + \hat{z}^2$ for all $z,\hat{z}>0$, we have:

$$
\begin{aligned}
\sum_{c\in\mathcal{C}}\sum_{e\in\mathcal{H}}\vartheta_t(c,e)\nu_t(c,e)^2 &= \sum_{c\in\mathcal{C}}\sum_{e\in\mathcal{H}}\vartheta_t(c,e)\left(\sum_{x\in\mathcal{X}}\lambda_t(c,x,e(x))\right)^2 \\
&= \sum_{c\in\mathcal{C}}\sum_{e\in\mathcal{H}}\vartheta_t(c,e)\left(\sum_{x\in\mathcal{X}}\sum_{a\in\mathcal{A}}[\![e(x)=a]\!]\lambda_t(c,x,a)\right)^2 \\
&\le 2\sum_{c\in\mathcal{C}}\sum_{e\in\mathcal{H}}\vartheta_t(c,e)\left(\sum_{x\in\mathcal{X}}\sum_{a\in\mathcal{A}}[\![e(x)=a]\!][\![\beta_t\le 1]\!][\![(c,x,a)=(c_t,x_t,a_t)]\!]\frac{\ell_t(a_t)}{\pi_t(c_t,x_t,a_t)}\right)^2 \\
&\quad + 2\sum_{c\in\mathcal{C}}\sum_{e\in\mathcal{H}}\vartheta_t(c,e)\left(\sum_{x\in\mathcal{X}}\sum_{a\in\mathcal{A}}[\![e(x)=a]\!][\![c=\tilde{\kappa}_t(\xi_t,a)]\!]\mu_t(c,x)\right)^2 \\
&\quad + 2\sum_{c\in\mathcal{C}}\sum_{e\in\mathcal{H}}\vartheta_t(c,e)\left(\sum_{x\in\mathcal{X}}\sum_{a\in\mathcal{A}}[\![e(x)=a]\!][\![c=\tilde{\kappa}'_t(\xi_t,a)]\!]\mu_t(c,x)\right)^2
\end{aligned}
$$
$$\tag{15}$$

Note that:

$$\sum_{c\in\mathcal{C}}\sum_{e\in\mathcal{H}}\vartheta_t(c,e)\left(\sum_{x\in\mathcal{X}}\sum_{a\in\mathcal{A}}[\![e(x)=a]\!][\![c=\tilde{\kappa}_t(\xi_t,a)]\!]\mu_t(c,x)\right)^2$$

$$=\sum_{c\in\mathcal{C}}\sum_{e\in\mathcal{H}}\vartheta_t(c,e)\left(\sum_{x\in\mathcal{X}}\sum_{a\in\mathcal{A}}[\![e(x)=a]\!][\![c=\tilde{\kappa}_t(\xi_t,a)]\!]\mu_t(c,x)\right)\left(\sum_{x\in\mathcal{X}}\sum_{a\in\mathcal{A}}[\![e(x)=a]\!][\![c=\tilde{\kappa}_t(\xi_t,a)]\!]\mu_t(c,x)\right)$$

$$\leq\sum_{c\in\mathcal{C}}\sum_{e\in\mathcal{H}}\vartheta_t(c,e)\left(\sum_{x\in\mathcal{X}}\sum_{a\in\mathcal{A}}[\![c=\tilde{\kappa}_t(\xi_t,a)]\!]\mu_t(c,x)\right)\left(\sum_{x\in\mathcal{X}}\mu_t(c,x)\sum_{a\in\mathcal{A}}[\![e(x)=a]\!]\right)$$

$$=\sum_{c\in\mathcal{C}}\sum_{e\in\mathcal{H}}\vartheta_t(c,e)\left(\sum_{a\in\mathcal{A}}[\![c=\tilde{\kappa}_t(\xi_t,a)]\!]\sum_{x\in\mathcal{X}}\mu_t(c,x)\right)\left(\sum_{x\in\mathcal{X}}\mu_t(c,x)\right)$$

$$=\sum_{c\in\mathcal{C}}\sum_{e\in\mathcal{H}}\vartheta_t(c,e)\sum_{a\in\mathcal{A}}[\![c=\tilde{\kappa}_t(\xi_t,a)]\!]$$

$$=\sum_{a\in\mathcal{A}}\sum_{c\in\mathcal{C}}[\![c=\tilde{\kappa}_t(\xi_t,a)]\!]\sum_{e\in\mathcal{H}}\vartheta_t(c,e)$$

$$=\sum_{a\in\mathcal{A}}\sum_{c\in\mathcal{C}}[\![c=\tilde{\kappa}_t(\xi_t,a)]\!]$$

$$=\sum_{a\in\mathcal{A}}1$$

$$=K \tag{16}$$

Similarly we have:

$$\sum_{c\in\mathcal{C}}\sum_{e\in\mathcal{H}}\vartheta_t(c,e)\left(\sum_{x\in\mathcal{X}}\sum_{a\in\mathcal{A}}[\![e(x)=a]\!][\![c=\tilde{\kappa}'_t(\xi_t,a)]\!]\mu_t(c,x)\right)^2=K \tag{17}$$

Now note that, by Lemma B.4, we have, for all $a\in\mathcal{A}$ that:

$$\frac{[\![\beta_t\leq 1]\!]}{\pi_t(c_t,x_t,a)}\leq\frac{2}{\xi_t(c_t,x_t,a)}$$

so by Lemma B.1 we have:

$$\mathbb{E}\left[\sum_{c\in\mathcal{C}}\sum_{e\in\mathcal{H}}\vartheta_t(c,e)\left(\sum_{x\in\mathcal{X}}\sum_{a\in\mathcal{A}}[\![e(x)=a]\!][\![\beta_t\leq 1]\!][\![(c,x,a)=(c_t,x_t,a_t)]\!]\frac{\ell_t(a_t)}{\pi_t(c_t,x_t,a_t)}\right)^2\,\bigg|\,\vartheta_t\right]$$

$$=\mathbb{E}\left[\sum_{e\in\mathcal{H}}\vartheta_t(c_t,e)\left([\![e(x_t)=a_t]\!][\![\beta_t\leq 1]\!]\frac{\ell_t(a_t)}{\pi_t(c_t,x_t,a_t)}\right)^2\,\bigg|\,\vartheta_t\right]$$

$$=\sum_{a\in\mathcal{A}}\mathbb{P}[a_t=a\,|\,\vartheta_t]\sum_{e\in\mathcal{H}}\vartheta_t(c_t,e)\left([\![e(x_t)=a]\!][\![\beta_t\leq 1]\!]\frac{\ell_t(a)}{\pi_t(c_t,x_t,a)}\right)^2$$

$$=\sum_{a\in\mathcal{A}}\pi_t(c_t,x_t,a)\sum_{e\in\mathcal{H}}\vartheta_t(c_t,e)\left([\![e(x_t)=a]\!][\![\beta_t\leq 1]\!]\frac{\ell_t(a)}{\pi_t(c_t,x_t,a)}\right)^2$$

$$=\sum_{a\in\mathcal{A}}\sum_{e\in\mathcal{H}}\vartheta_t(c_t,e)[\![e(x_t)=a]\!][\![\beta_t\leq 1]\!]\frac{\ell_t(a)^2}{\pi_t(c_t,x_t,a)}$$

$$\leq\sum_{a\in\mathcal{A}}\sum_{e\in\mathcal{H}}\vartheta_t(c_t,e)[\![e(x_t)=a]\!]\frac{[\![\beta_t\leq 1]\!]}{\pi_t(c_t,x_t,a)}$$

$$\leq\sum_{a\in\mathcal{A}}\frac{2}{\xi_t(c_t,x_t,a)}\sum_{e\in\mathcal{H}}\vartheta_t(c_t,e)[\![e(x_t)=a]\!]$$

$$\leq\sum_{a\in\mathcal{A}}\frac{2}{\xi_t(c_t,x_t,a)}\xi_t(c_t,x_t,a)$$

$$\leq\sum_{a\in\mathcal{A}}2$$

$$=2K \tag{18}$$

Taking expectations on Equation (15) and then substituting in equations (16), (17) and (18) gives us the result.

## C.13  Lemma B.13

Take any $(c,e)\in\mathcal{C}\times\mathcal{H}$. Note first that for all $(x,a)\in\mathcal{X}\times\mathcal{A}$ we have:

$$\lambda_t(c,x,a)\geq -[\![c=\tilde{\kappa}'_t(\xi_t,a)]\!]\mu_t(c,x)\geq -\mu_t(c,x)$$

so that:

$$\nu_t(c,e)=\sum_{x\in\mathcal{X}}\lambda_t(c,x,e(x))$$

$$\geq\sum_{a\in\mathcal{A}}\sum_{x\in\mathcal{X}}\lambda_t(c,x,a)$$

$$\geq -\sum_{a\in\mathcal{A}}\sum_{x\in\mathcal{X}}\mu_t(c,x)$$

$$=-\sum_{a\in\mathcal{A}}1$$

$$\geq -K$$

as required.

## C.14  Lemma B.14

By the description of the UPDATE subroutine in Section **??** we see that our update procedure is equivalent to creating a function $\vartheta'_t\in\mathbb{R}^{\mathcal{C}\times\mathcal{H}}$ defined such that for all $(c,e)\in\mathcal{C}\times\mathcal{H}$ we have:

$$\vartheta'_t(c,e)=\vartheta_t(c,e)\prod_{x\in\mathcal{G}_t(c)}\exp(-\hat{\eta}\lambda_t(c,x,e(x)))$$

and then, for all $c \in \mathcal{C}$, normalising $\vartheta'_t(c, \circ)$ to create $\vartheta_t(c, \circ)$. Note that for all $(c, x) \in \mathcal{I}$ with $x \notin \mathcal{G}_t(c)$ we have $\mu_t(c, x) = 0$ and $(c, x) \neq (c_t, x_t)$ so that $\lambda_t(c, x, a) = 0$ for all $a \in \mathcal{A}$. This implies that for all $(c, e) \in \mathcal{C} \times \mathcal{H}$ we have:

$$\vartheta'_t(c, e) = \vartheta_t(c, e) \prod_{x \in \mathcal{X}} \exp(-\hat{\eta} \lambda_t(c, x, e(x))) = \vartheta_t(c, e) \exp\left(-\hat{\eta} \sum_{x \in \mathcal{X}} \lambda_t(c, x, e(x))\right) = \vartheta_t(c, e) \exp(-\hat{\eta} \nu_t(c, e))$$

so that:

$$\vartheta_{t+1}(c, e) = \frac{\vartheta_t(c, e) \exp(-\hat{\eta} \nu_t(c, e))}{\sum_{e' \in \mathcal{H}} \vartheta_t(c, e') \exp(-\hat{\eta} \nu_t(c, e'))}$$

as required.

### C.15 Lemma B.15

Take any $c \in \mathcal{C}$. Define:

$$z := \sum_{e \in \mathcal{H}} \vartheta_t(c, e) \exp(-\hat{\eta} \nu_t(c, e))$$

By Lemma B.14 we have:

$$
\begin{aligned}
B(\vartheta^*, \vartheta_t(c, \circ)) - B(\vartheta^*, \vartheta_{t+1}(c, \circ)) &= \sum_{e \in \mathcal{H}} \vartheta^*(c, e) \left( \ln\left(\frac{\vartheta^*(c, e)}{\vartheta_t(c, e)}\right) - \ln\left(\frac{\vartheta^*(c, e)}{\vartheta_{t+1}(c, e)}\right) \right) \\
&= \sum_{e \in \mathcal{H}} \vartheta^*(c, e) \ln\left(\frac{\vartheta_{t+1}(c, e)}{\vartheta_t(c, e)}\right) \\
&= \sum_{e \in \mathcal{H}} \vartheta^*(c, e) \ln\left(\frac{\exp(-\hat{\eta} \nu_t(c, e))}{z}\right) \\
&= -\hat{\eta} \sum_{e \in \mathcal{H}} \vartheta^*(c, e) \nu_t(c, e) - \ln(z) \sum_{e \in \mathcal{H}} \vartheta^*(c, e) \\
&= -\hat{\eta} \sum_{e \in \mathcal{H}} \vartheta^*(c, e) \nu_t(c, e) - \ln(z) \qquad (19)
\end{aligned}
$$

Since $\eta \leq \sqrt{T/K}$ we have $\hat{\eta} \leq 1/K$ so that, by Lemma B.13, we have, for all $e \in \mathcal{H}$ that:

$$\hat{\eta} \nu_t(c, e) \geq -1$$

so, since $\exp(-\hat{z}) \leq 1 - \hat{z} + \hat{z}^2$ for all $x \geq -1$ and $\ln(1 + \hat{z}) \leq \hat{z}$ for all $\hat{z} \in \mathbb{R}$, we have:

$$
\begin{aligned}
\ln(z) &= \ln\left(\sum_{e \in \mathcal{H}} \vartheta_t(c, e) \exp(-\hat{\eta} \nu_t(c, e))\right) \\
&\leq \ln\left(\sum_{e \in \mathcal{H}} \vartheta_t(c, e)(1 - \hat{\eta} \nu_t(c, e) + \hat{\eta}^2 \nu_t(c, e)^2)\right) \\
&= \ln\left(\sum_{e \in \mathcal{H}} \vartheta_t(c, e) + \sum_{e \in \mathcal{H}} \vartheta_t(c, e)(-\hat{\eta} \nu_t(c, e) + \hat{\eta}^2 \nu_t(c, e)^2)\right) \\
&= \ln\left(1 + \sum_{e \in \mathcal{H}} \vartheta_t(c, e)(-\hat{\eta} \nu_t(c, e) + \hat{\eta}^2 \nu_t(c, e)^2)\right) \\
&\leq \sum_{e \in \mathcal{H}} \vartheta_t(c, e)(-\hat{\eta} \nu_t(c, e) + \hat{\eta}^2 \nu_t(c, e)^2) \\
&= -\hat{\eta} \sum_{e \in \mathcal{H}} \vartheta_t(c, e) \nu_t(c, e) + \hat{\eta}^2 \sum_{e \in \mathcal{H}} \vartheta_t(c, e) \nu_t(c, e)^2 \qquad (20)
\end{aligned}
$$

Substituting Equation (20) into Equation (19) gives us the result.

### C.16  Lemma B.16

Immediate from summing the inequality in Lemma B.15 over all $c \in \mathcal{C}$, taking expectations (conditioned on $\vartheta_t$) and substituting in the inequalities in lemmas B.11 and B.12.

### C.17  Lemma B.17

From Lemma B.16 we have, for all $t \in [T]$, that:

$$\hat{\eta}\mathbb{E}[r_t] - 8\hat{\eta}^2 K = \mathbb{E}\left[\sum_{c \in \mathcal{C}}(B(\vartheta^*, \vartheta_t(c, \circ)) - B(\vartheta^*, \vartheta_{t+1}(c, \circ)))\right]$$

and hence, by summing over all $t \in [T]$ and noting the linearity of expectation and the fact that the relative entropy is non-negative, we have:

$$\hat{\eta}\sum_{t \in [T]}\mathbb{E}[r_t] - 8\hat{\eta}^2 KT = \mathbb{E}\left[\sum_{c \in \mathcal{C}}\sum_{t \in [T]}(B(\vartheta^*, \vartheta_t(c, \circ)) - B(\vartheta^*, \vartheta_{t+1}(c, \circ)))\right]$$

$$= \mathbb{E}\left[\sum_{c \in \mathcal{C}}(B(\vartheta^*, \vartheta_1(c, \circ)) - B(\vartheta^*, \vartheta_{T+1}(c, \circ)))\right]$$

$$\leq \mathbb{E}\left[\sum_{c \in \mathcal{C}}B(\vartheta^*, \vartheta_1(c, \circ))\right]$$

$$= \sum_{c \in \mathcal{C}}B(\vartheta^*, \vartheta)$$

so that:

$$\sum_{t \in [T]}\mathbb{E}[r_t] \leq \frac{1}{\hat{\eta}}\sum_{c \in \mathcal{C}}B(\vartheta^*, \vartheta) + 8\hat{\eta}KT$$

noting that $\hat{\eta} = \eta/\sqrt{KT}$, we have the result.

## D  Details of Examples

We now describe and analyse the instances of SPEW required to obtain the results in Section 5. We start with the following lemma:

**Lemma D.1.** *Given a tree $\mathcal{T}$ with edge set $\mathcal{E}$ and any $\pi : \mathcal{T} \times \mathcal{A} \to [0, 1]$ with $\pi(v, \circ) \in \Delta_{\mathcal{A}}$ for all $v \in \mathcal{T}$, there exists a distribution $\theta \in \Delta_{\mathcal{A}^{\mathcal{T}}}$ such that:*

$$\pi(v, a) = \sum_{e \in \mathcal{A}^{\mathcal{T}}}[\![e(v) = a]\!]\theta(e) \quad \forall (v, a) \in \mathcal{T} \times \mathcal{A}$$

*and:*

$$\sum_{e \in \mathcal{A}^{\mathcal{T}}}\theta(e)\sum_{(u,v) \in \mathcal{E}}[\![e(u) \neq e(v)]\!] = \frac{1}{2}\sum_{(u,v) \in \mathcal{E}}\sum_{a \in \mathcal{A}}|\pi(u, a) - \pi(v, a)|$$

*Proof.* We prove by induction on $|\mathcal{T}|$. The inductive hypothesis clearly holds for $|\mathcal{T}| = 1$. Now consider a tree $\mathcal{T}$ with $|\mathcal{T}| > 1$ and let $\lambda$ be a leaf of $\mathcal{T}$. Assume that the inductive hypothesis holds for the tree $\mathcal{T}'$ created by removing $\lambda$ from $\mathcal{T}$. By now proving that the inductive hypothesis holds for $\mathcal{T}$ we will have completed the proof of the lemma.

Let $\lambda'$ be the neighbour of $\lambda$. For all $a \in \mathcal{A}$ let $\xi(a) := \pi(\lambda, a)$ and $\xi'(a) := \pi(\lambda', a)$. Let $\mathcal{S}$ be the set of all $a \in \mathcal{A}$ with $\xi(a) \leq \xi'(a)$. For all $a \in \mathcal{S}$ let:

$$r(a) := \frac{\xi(a)}{\xi'(a)}$$

for all $a \in \mathcal{A} \setminus \mathcal{S}$ let
$$q(a) := \frac{\xi(a) - \xi'(a)}{\sum_{a' \in \mathcal{A} \setminus \mathcal{S}}(\xi(a') - \xi'(a'))}$$

For all $e \in \mathcal{A}^{\mathcal{T}'}$ and $a \in \mathcal{A}$ let $\gamma(e, a)$ be the function $e' \in \mathcal{A}^{\mathcal{T}}$ with $e'(\lambda) = a$ and $e'(v) = e(v)$ for all $v \in \mathcal{T}'$. Let $\mathcal{E}'$ be the edge set of $\mathcal{T}'$. By the inductive hypothesis, let $\theta' \in \Delta_{\mathcal{A}^{\mathcal{T}'}}$ be such that:
$$\pi(v, a) = \sum_{e \in \mathcal{A}^{\mathcal{T}'}} [\![e(v) = a]\!]\theta'(e) \quad \forall (v, a) \in \mathcal{T}' \times \mathcal{A}$$

and:
$$\sum_{e \in \mathcal{A}^{\mathcal{T}'}} \theta'(e) \sum_{(u,v) \in \mathcal{E}'} [\![e(u) \neq e(v)]\!] = \frac{1}{2} \sum_{(u,v) \in \mathcal{E}'} \sum_{a \in \mathcal{A}} |\pi(u, a) - \pi(v, a)|$$

For all $a \in \mathcal{A} \setminus \mathcal{S}$ and $e \in \mathcal{A}^{\mathcal{T}'}$ with $e(\lambda') = a$ define:
$$\theta(\gamma(e, a)) := \theta'(e)$$

For all $a \in \mathcal{S}$ and $e \in \mathcal{A}^{\mathcal{T}'}$ with $e(\lambda') = a$ define:
$$\theta(\gamma(e, a)) := r(a)\theta'(e)$$

For all $a \in \mathcal{S}$, $a' \in \mathcal{A} \setminus \mathcal{S}$ and $e \in \mathcal{A}^{\mathcal{T}'}$ with $e(\lambda') = a$ define:
$$\theta(\gamma(e, a')) := q(a')(1 - r(a))\theta'(e)$$

For all other $e' \in \mathcal{A}^{\mathcal{T}}$ we define $\theta(e') := 0$.

Note that for all $v \in \mathcal{T}'$ and $a \in \mathcal{A}$ we have:
$$\pi(v, a) = \sum_{e \in \mathcal{A}^{\mathcal{T}'}} [\![e(v) = a]\!]\theta'(e) = \sum_{e \in \mathcal{A}^{\mathcal{T}'}} [\![e(v) = a]\!] \sum_{a' \in \mathcal{A}} \theta(\gamma(e, a')) = \sum_{e' \in \mathcal{A}^{\mathcal{T}}} [\![e'(v) = a]\!]\theta(e')$$

and for all $a \in \mathcal{S}$ we have:
$$\pi(\lambda, a) = \xi(a) = r(a)\xi'(a) = r(a)\pi(\lambda', a) = r(a) \sum_{e \in \mathcal{A}^{\mathcal{T}'}} [\![e(\lambda') = a]\!]\theta'(e)$$
$$= \sum_{e \in \mathcal{A}^{\mathcal{T}'}} [\![e(\lambda') = a]\!]\theta(\gamma(e, a)) = \sum_{e' \in \mathcal{A}^{\mathcal{T}}} [\![e'(\lambda) = a]\!]\theta(e')$$

and for all $a \in \mathcal{A} \setminus \mathcal{S}$ we have:
$$\pi(\lambda, a) = \xi(a)$$
$$= \xi'(a) + (\xi(a) - \xi'(a))$$
$$= \xi'(a) + q(a) \sum_{a' \in \mathcal{A} \setminus \mathcal{S}} (\xi(a') - \xi'(a'))$$
$$= \xi'(a) + q(a) \sum_{a' \in \mathcal{S}} (\xi'(a') - \xi(a'))$$
$$= \xi'(a) + q(a) \sum_{a' \in \mathcal{S}} (1 - r(a'))\xi'(a')$$
$$= \pi(\lambda', a) + q(a) \sum_{a' \in \mathcal{S}} (1 - r(a'))\pi(\lambda', a')$$
$$= \sum_{e \in \mathcal{A}^{\mathcal{T}'}} [\![e(\lambda') = a]\!]\theta'(e) + q(a) \sum_{a' \in \mathcal{S}} (1 - r(a')) \sum_{e \in \mathcal{A}^{\mathcal{T}'}} [\![e(\lambda') = a']\!]\theta'(e)$$
$$= \sum_{e \in \mathcal{A}^{\mathcal{T}'}} [\![e(\lambda') = a]\!]\theta'(e) + \sum_{a' \in \mathcal{S}} \sum_{e \in \mathcal{A}^{\mathcal{T}'}} [\![e(\lambda') = a']\!]q(a)(1 - r(a'))\theta'(e)$$
$$= \sum_{e \in \mathcal{A}^{\mathcal{T}'}} [\![e(\lambda') = a]\!]\theta(\gamma(e, a)) + \sum_{a' \in \mathcal{S}} \sum_{e \in \mathcal{A}^{\mathcal{T}'}} [\![e(\lambda') = a']\!]\theta(\gamma(e, a))$$
$$= \sum_{e' \in \mathcal{A}^{\mathcal{T}}} [\![e'(\lambda) = a]\!]\theta(e')$$

We have now shown that for all $v \in \mathcal{T}$ and $a \in \mathcal{A}$ we have:

$$\pi(v, a) = \sum_{e \in \mathcal{A}^{\mathcal{T}}} [\![e(v) = a]\!] \theta(e)$$

as required.

For all $e \in \mathcal{A}^{\mathcal{T}}$ define:

$$\Lambda(e) := \sum_{(u,v) \in \mathcal{E}} [\![e(u) \neq e(v)]\!]$$

and for all $e \in \mathcal{A}^{\mathcal{T}'}$ define:

$$\Lambda(e) := \sum_{(u,v) \in \mathcal{E}'} [\![e(u) \neq e(v)]\!]$$

Note that for all $a \in \mathcal{A} \setminus \mathcal{S}$ and $e \in \mathcal{A}^{\mathcal{T}'}$ with $e(\lambda') = a$ we have:

$$\sum_{a' \in \mathcal{A}} \theta(\gamma(e, a')) \Lambda(\gamma(e, a')) = \theta(\gamma(e, a)) \Lambda(\gamma(e, a)) = \theta'(e) \Lambda(e)$$

and for all $a \in \mathcal{S}$ and $e \in \mathcal{A}^{\mathcal{T}'}$ with $e(\lambda') = a$ we have:

$$\sum_{a' \in \mathcal{A}} \theta(\gamma(e, a')) \Lambda(\gamma(e, a')) = \theta(\gamma(e, a)) \Lambda(\gamma(e, a)) + \sum_{a' \in \mathcal{A} \setminus \mathcal{S}} \theta(\gamma(e, a')) \Lambda(\gamma(e, a'))$$

$$= \theta(\gamma(e, a)) \Lambda(e) + (\Lambda(e) + 1) \sum_{a' \in \mathcal{A} \setminus \mathcal{S}} \theta(\gamma(e, a'))$$

$$= r(a) \theta'(e) \Lambda(e) + (\Lambda(e) + 1)(1 - r(a)) \theta'(e)$$

$$= \theta'(e) \Lambda(e) + (1 - r(a)) \theta'(e)$$

Hence, we have:

$$\sum_{e' \in \mathcal{A}^{\mathcal{T}}} \theta(e') \Lambda(e') = \sum_{e \in \mathcal{A}^{\mathcal{T}'}} \theta'(e) \Lambda(e) + \sum_{a \in \mathcal{S}} (1 - r(a)) \sum_{e \in \mathcal{A}^{\mathcal{T}'}} [\![e(\lambda') = a]\!] \theta'(e)$$

$$= \sum_{e \in \mathcal{A}^{\mathcal{T}'}} \theta'(e) \Lambda(e) + \sum_{a \in \mathcal{S}} (1 - r(a)) \pi(\lambda', a)$$

$$= \sum_{e \in \mathcal{A}^{\mathcal{T}'}} \theta'(e) \Lambda(e) + \sum_{a \in \mathcal{S}} (1 - r(a)) \xi'(a)$$

$$= \sum_{e \in \mathcal{A}^{\mathcal{T}'}} \theta'(e) \Lambda(e) + \sum_{a \in \mathcal{S}} (\xi'(a) - \xi(a))$$

$$= \sum_{e \in \mathcal{A}^{\mathcal{T}'}} \theta'(e) \Lambda(e) + \frac{1}{2} \sum_{a \in \mathcal{A}} |\pi(\lambda, a) - \pi(\lambda', a)|$$

$$= \frac{1}{2} \sum_{(u,v) \in \mathcal{E}} \sum_{a \in \mathcal{A}} |\pi(u, a) - \pi(v, a)|$$

as required.

$\square$

### D.1   Trees with a known distribution

Here we define $\mathcal{X} := \mathcal{V}$ and $\chi_t(c, v) := v$ for all $t \in [T]$ and $(c, v) \in \mathcal{C} \times \mathcal{V}$. We define $\vartheta$ by:

$$\vartheta(e) := \frac{1}{K} \prod_{(u,v) \in \mathcal{E}} \left( [\![e(u) = e(v)]\!] \left(1 - \frac{1}{N}\right) + [\![e(u) \neq e(v)]\!] \frac{1}{(K-1)N} \right) \quad \forall e \in \mathcal{H}$$

This instance of HEDGE can be implemented by *belief propagation* [29], which gives us the time and space complexity bounds.

For all $t \in [T]$, we define the target $\mu_t$ by:

$$\mu_t(c,x) := \frac{\rho(c,x)}{\sum_{v \in \mathcal{X}} \rho(c,v)} \quad \forall (c,x) \in \mathcal{C} \times \mathcal{X}$$

To analyse the regret, suppose we have a policy $\tilde{\pi} \in \mathcal{Q}$ which has statistical parity with respect to $\rho$, noting that:

$$\tilde{\pi} \in \bigcap_{t \in [T]} \mathcal{F}(\mu_t)\,.$$

By Lemma D.1 we have $\vartheta^* : \mathcal{C} \times \mathcal{H} \to [0,1]$ with:

$$\tilde{\pi}(c,x,a) = \sum_{e \in \mathcal{H}} [\![e(x) = a]\!] \vartheta^*(c,e) \quad \forall (c,x,a) \in \mathcal{C} \times \mathcal{X} \times \mathcal{A}$$

and:

$$\sum_{e \in \mathcal{H}} \vartheta^*(c,e) \sum_{(u,v) \in \mathcal{E}} [\![e(u) \neq e(v)]\!] = \frac{1}{2} \sum_{(u,v) \in \mathcal{E}} \sum_{a \in \mathcal{A}} |\tilde{\pi}(c,u,a) - \tilde{\pi}(c,v,a)| \quad \forall c \in \mathcal{C}$$

We will now bound the term $\Phi$ that appears in Theorem 4.1 using our choice of $\tilde{\pi}$ and $\vartheta^*$, which will give us the result. To do this note that for all $e \in \mathcal{H}$ we have:

$$-\ln(\vartheta(e)) \in \mathcal{O}\left( \ln(K) + \ln(KN) \sum_{(u,v) \in \mathcal{E}} [\![e(u) \neq e(v)]\!] \right)$$

so that:

$$\Phi = \sum_{c \in \mathcal{C}} \sum_{e \in \mathcal{H}} \vartheta^*(c,e) \ln\left( \frac{\vartheta^*(c,e)}{\vartheta(e)} \right)$$

$$\leq - \sum_{c \in \mathcal{C}} \sum_{e \in \mathcal{H}} \vartheta^*(c,e) \ln(\vartheta(e))$$

$$\in \mathcal{O}\left( \ln(K) \sum_{c \in \mathcal{C}} \sum_{e \in \mathcal{H}} \vartheta^*(c,e) + \ln(KN) \sum_{c \in \mathcal{C}} \sum_{e \in \mathcal{H}} \vartheta^*(c,e) \sum_{(u,v) \in \mathcal{E}} [\![e(u) \neq e(v)]\!] \right)$$

$$= \mathcal{O}\left( \ln(K) M + \ln(KN) \sum_{c \in \mathcal{C}} \sum_{(u,v) \in \mathcal{E}} \sum_{a \in \mathcal{A}} |\tilde{\pi}(c,u,a) - \tilde{\pi}(c,v,a)| \right)$$

$$= \mathcal{O}(\ln(K) M + \ln(KN) \Psi(\tilde{\pi}))$$

### D.2 Empirical Statistical Parity

Here we take $\mathcal{X} := \mathcal{V} \times [T]$ and define, for all $t \in [T]$ and $(c,v) \in \mathcal{C} \times \mathcal{V}$, $\chi_t(c,v) := (v,t)$ and:

$$\mathcal{G}_t(c) := \{(v,t) \mid v \in \mathcal{V}\}$$

We then define $\vartheta$ by:

$$\vartheta(e) := \prod_{v \in \mathcal{V}} \frac{1}{K} \prod_{t \in [T-1]} \left( [\![e(v,t) = e(v,t+1)]\!] \left( 1 - \frac{1}{T} \right) + [\![e(v,t) \neq e(v,t+1)]\!] \frac{1}{(K-1)T} \right)$$

for all $e \in \mathcal{H}$. This instance of HEDGE is implemented by running the FIXEDSHARE algorithm of [16] for each context, which gives the time and space complexity.

For all $t \in [T]$, we define:

$$\mu_t(c,(v,t)) := \frac{\rho_t(c,v)}{\sum_{v' \in \mathcal{V}} \rho_t(c,v')} \quad \forall (c,v) \in \mathcal{C} \times \mathcal{V}$$

and:

$$\mu_t(c, (v, s)) := 0 \quad \forall (c, v, s) \in \mathcal{C} \times \mathcal{V} \times ([T \setminus \{t\}]$$

Note that if we have yet to see instances with a particular protected characteristic $c \in \mathcal{C}$ then $\mu_t(c, (v, t))$ is undefined for all $v \in \mathcal{V}$. It is straightforward to modify the algorithm to handle such situations.

We now analyse the regret. Suppose we have a sequence of policies $\tilde{\boldsymbol{\pi}} \in \mathcal{Q}^T$ such that for all $t \in [T]$ we have that $\tilde{\pi}_t$ has statistical parity with respect to $\rho_t$. Define the virtual policy $\tilde{\pi}' \in \mathcal{P}$ such that:

$$\tilde{\pi}'(c, (v, t), \circ) := \tilde{\pi}_t(c, v, \circ) \quad \forall (c, v, t) \in \mathcal{C} \times \mathcal{V} \times [T]$$

Note that we have $\tilde{\pi}' \in \bigcap \mathcal{F}(\mu_t)$ as required.

Define the set $\mathcal{H}' = \mathcal{A}^{[T]}$. By Lemma D.1 (with $\mathcal{T}$ being a chain), for all $(c, v) \in \mathcal{C} \times \mathcal{V}$ we can construct a distribution $\vartheta'(c, v, \circ) \in \Delta_{\mathcal{H}'}$ in which:

$$\tilde{\pi}'(c, (v, t), a) = \sum_{e' \in \mathcal{H}'} [\![ e'(t) = a ]\!] \vartheta'(c, v, e')$$

and:

$$\sum_{e' \in \mathcal{H}'} \vartheta'(c, v, e') \sum_{t \in [T-1]} [\![ e'(t) \neq e'(t+1) ]\!] = \frac{1}{2} \sum_{t \in [T-1]} \sum_{a \in \mathcal{A}} |\tilde{\pi}'(c, (v, t), a) - \tilde{\pi}'(c, (v, t+1), a)|$$

Now define, for all $c \in \mathcal{C}$, the distribution $\vartheta^*(c, \circ) \in \Delta_{\mathcal{H}}$ by:

$$\vartheta^*(c, e) = \prod_{v \in \mathcal{V}} \vartheta'(c, v, e(v, \circ)) \quad \forall e \in \mathcal{H}$$

Since for each expert $e \in \mathcal{H}$ we have:

$$-\ln(\vartheta(e)) \in \mathcal{O} \left( \ln(K)N + \ln(KT) \sum_{v \in \mathcal{V}} \sum_{t \in [T-1]} [\![ e(v, t) \neq e(v, t+1) ]\!] \right)$$

we also have, for each $c \in \mathcal{C}$, that:

$$\sum_{e \in \mathcal{H}} \vartheta^*(c, e) \ln\left(\frac{\vartheta^*(c, e)}{\vartheta(e)}\right)$$

$$\leq -\sum_{e \in \mathcal{H}} \vartheta^*(c, e) \ln(\vartheta(e))$$

$$\in \mathcal{O}\left(\sum_{e \in \mathcal{H}} \vartheta^*(c, e) \left(\ln(K)N + \ln(KT) \sum_{v \in \mathcal{V}} \sum_{t \in [T-1]} [\![e(v, t) \neq e(v, t+1)]\!]\right)\right)$$

$$= \mathcal{O}\left(\ln(K)N + \ln(KT) \sum_{e \in \mathcal{H}} \vartheta^*(c, e) \sum_{v \in \mathcal{V}} \sum_{t \in [T-1]} [\![e(v, t) \neq e(v, t+1)]\!]\right)$$

$$= \mathcal{O}\left(\ln(K)N + \ln(KT) \sum_{e \in \mathcal{H}} \prod_{u \in \mathcal{V}} \vartheta'(c, u, e(u, \circ)) \sum_{v \in \mathcal{V}} \sum_{t \in [T-1]} [\![e(v, t) \neq e(v, t+1)]\!]\right)$$

$$= \mathcal{O}\left(\ln(K)N + \ln(KT) \sum_{v \in \mathcal{V}} \sum_{e \in \mathcal{H}} \prod_{u \in \mathcal{V}} \vartheta'(c, u, e(u, \circ)) \sum_{t \in [T-1]} [\![e(v, t) \neq e(v, t+1)]\!]\right)$$

$$= \mathcal{O}\left(\ln(K)N + \ln(KT) \sum_{v \in \mathcal{V}} \sum_{e(v, \circ) \in \mathcal{H}'} \vartheta'(c, v, e(v, \circ)) \sum_{t \in [T-1]} [\![e(v, t) \neq e(v, t+1)]\!]\right)$$

$$= \mathcal{O}\left(\ln(K)N + \ln(KT) \sum_{v \in \mathcal{V}} \sum_{e' \in \mathcal{H}'} \vartheta'(c, v, e') \sum_{t \in [T-1]} [\![e'(t) \neq e'(t+1)]\!]\right)$$

$$= \mathcal{O}\left(\ln(K)N + \ln(KT) \sum_{v \in \mathcal{V}} \sum_{t \in [T-1]} \sum_{a \in \mathcal{A}} |\tilde{\pi}'(c, (v, t), a) - \tilde{\pi}'(c, (v, t+1), a)|\right)$$

$$= \mathcal{O}\left(\ln(K)N + \ln(KT) \sum_{v \in \mathcal{V}} \sum_{t \in [T-1]} \sum_{a \in \mathcal{A}} |\tilde{\pi}_t(c, v, a) - \tilde{\pi}_{t+1}(c, v, a)|\right)$$

so that:

$$\sum_{c \in \mathcal{C}} \sum_{e \in \mathcal{H}} \vartheta^*(c, e) \ln\left(\frac{\vartheta^*(c, e)}{\vartheta(e)}\right) \in \mathcal{O}(MN + \Psi(\tilde{\boldsymbol{\pi}})) \ln(KT)$$

We also have:

$$\tilde{\pi}'(c, (v, t), a) = \sum_{e(v, \circ) \in \mathcal{H}'} [\![e(v, t) = a]\!] \vartheta'(c, v, e(v, \circ)) = \sum_{e \in \mathcal{H}} [\![e(v, t) = a]\!] \vartheta^*(c, e)$$

for all $(c, v, t, a) \in \mathcal{C} \times \mathcal{V} \times [T] \times \mathcal{A}$.

Substituting into Theorem 4.1 gives us the regret bound.

### D.3 Approximate statistical parity for hierarchical clusterings with an unknown distribution

Here we take $\mathcal{X}$ to be the set of sets in the tree $\mathcal{T}$ that is the hierarchical clustering. Let $\mathcal{E}$ be the edge set of $\mathcal{T}$. The functions $\chi_t$ will be constructed online by the characteristic/context sets seen so far. We define $\vartheta$ by:

$$\vartheta(e) := \frac{1}{K} \prod_{(x, x') \in \mathcal{E}} \left([\![e(x) = e(x')]\!] \left(1 - \frac{1}{|\mathcal{T}|}\right) + [\![e(x) \neq e(x')]\!] \frac{1}{(K-1)|\mathcal{T}|}\right) \quad \forall e \in \mathcal{H}$$

Now define:

$$\hat{\epsilon} := \epsilon/4H \quad ; \quad n := \lceil \ln(2T/\delta)/2\hat{\epsilon}^2 \rceil$$

For each $c \in \mathcal{C}$ we maintain a dynamic (in that it grows over time) full binary subtree $\mathcal{S}(c) \subseteq \mathcal{T}$ which contains the root. $\mathcal{S}(c)$ is initialised to contain the root (which is the set $\mathcal{V}$) as a single vertex. Let $\mathcal{L}(c)$ be the set of leaves of $\mathcal{S}(c)$ so that $\mathcal{L}(c)$ is a partition of $\mathcal{V}$. On each trial $t \in [T]$ we define, for all $(c, v) \in \mathcal{C} \times \mathcal{V}$, $\chi_t(c, v)$ to be the unique set in $\mathcal{L}(c)$ that contains $v$, and define $\mathcal{G}_t(c)$ to be the set $\mathcal{L}(c)$ at trial $t$. The instance of HEDGE for characteristic $c$ is implemented by belief propagation over the tree $\mathcal{S}(c)$ as in [26]. This gives us the time and space complexity bounds.

We now describe how $\mathcal{S}(c)$ grows and how the functions $\mu_t$ are defined. To do this we will define, for each $c \in \mathcal{C}$, a function $\tilde{\mu}(c, \circ) : \mathcal{S}(c) \to [0, 1]$ in which $\tilde{\mu}(c, \mathcal{V}) := 1$. For each $c \in \mathcal{C}$, $\mathcal{S}(c)$ and $\tilde{\mu}(c, \circ)$ grow as follows. Whenever we have a non-singleton set in $\mathcal{Y} \in \mathcal{L}(c)$ in which there have been $n$ trials $t$ so far in which $c_t = c$ and $x_t \in \mathcal{Y}$ then we add the two children of $\mathcal{Y}$ to $\mathcal{S}(c)$. Denoting the two children by $\mathcal{Z}$ and $\mathcal{Z}'$ we let $m$ be the number of trials $t$ in which $c_t = c$ and $x_t \in \mathcal{Z}$ and then define:

$$\tilde{\mu}(c, \mathcal{Z}) := m\tilde{\mu}(c, \mathcal{Y})/n \quad ; \quad \tilde{\mu}(c, \mathcal{Z}') := (n - m)\tilde{\mu}(c, \mathcal{Y})/n$$

On each trial $t$ we define $\mu_t(c, \circ)$ as follows. For $x \in \mathcal{G}_t(c)$ we have $\mu_t(c, x) := \tilde{\mu}(c, x)$ and for $x \in \mathcal{X} \setminus \mathcal{G}_t(c)$ we have $\mu_t(c, x) := 0$. This completes the description of the instance of SPEW.

We now analyse this instance of SPEW. First, for all $(c, \mathcal{Y}) \in \mathcal{C} \times \mathcal{X}$ define:

$$\mu^*(c, \mathcal{Y}) := \frac{\sum_{v \in \mathcal{Y}} \rho^*(c, v)}{\sum_{v \in \mathcal{V}} \rho^*(c, v)}$$

For all $c \in \mathcal{C}$ let $\mathcal{S}^\dagger(c)$ be the tree $\mathcal{S}(c)$ on trial $T$ and let $\mathcal{L}^\dagger(c)$ be the set of leaves of $\mathcal{S}^\dagger(c)$. We define the distribution $\rho$ such that, for all $c \in \mathcal{C}$, $\mathcal{Y} \in \mathcal{L}^\dagger(c)$ and $v \in \mathcal{Y}$, we have:

$$\rho(c, v) := \frac{\tilde{\mu}(c, \mathcal{Y})\rho^*(c, v)}{\mu^*(c, \mathcal{Y})}$$

Note that by induction over time we always have:

$$\sum_{\mathcal{Y} \in \mathcal{L}(c)} \tilde{\mu}(c, \mathcal{Y}) = 1$$

so that:

$$
\begin{aligned}
\sum_{(c,v) \in \mathcal{C} \times \mathcal{V}} \rho(c, v) &= \sum_{c \in \mathcal{C}} \sum_{\mathcal{Y} \in \mathcal{L}^\dagger(c)} \sum_{v \in \mathcal{Y}} \frac{\tilde{\mu}(c, \mathcal{Y})\rho^*(c, v)}{\mu^*(c, \mathcal{Y})} \\
&= \sum_{c \in \mathcal{C}} \sum_{\mathcal{Y} \in \mathcal{L}^\dagger(c)} \tilde{\mu}(c, \mathcal{Y}) \sum_{v \in \mathcal{V}} \rho^*(c, v) \\
&= \sum_{c \in \mathcal{C}} \sum_{v \in \mathcal{V}} \rho^*(c, v) \sum_{\mathcal{Y} \in \mathcal{L}^\dagger(c)} \tilde{\mu}(c, \mathcal{Y}) \\
&= \sum_{c \in \mathcal{C}} \sum_{v \in \mathcal{V}} \rho^*(c, v) \\
&= 1
\end{aligned}
$$

and hence $\rho$ is a probability distribution as required.

We now show that with probability at least $1 - \delta$ we have that $\rho$ is $\epsilon$-close to $\rho^*$. Let $r$ be the root of $\mathcal{T}$, which is equal to the set $\mathcal{V}$. For any vertex $x \in \mathcal{T}$ let $\uparrow(x)$ be its parent and let $\triangleleft(x)$ and $\triangleright(x)$ be its left and right children respectively, if they exist. For all $c \in \mathcal{C}$ and $x \in \mathcal{S}^\dagger(c) \setminus \{r\}$ define:

$$p(c, x) := \frac{\tilde{\mu}(c, x)}{\tilde{\mu}(c, \uparrow(x))} \quad ; \quad p^*(c, x) := \frac{\mu^*(c, x)}{\mu^*(c, \uparrow(x))} \quad ; \quad \delta'(c, x) := p^*(c, x) - p(c, x)$$

Note that by the generation of $\tilde{\mu}(c, x)$ we have, direct from Hoeffding's inequality, that:

$$\mathbb{P}[|\delta'(c, x)| > \hat{\epsilon}] \leq 2 \exp(-2\hat{\epsilon}^2 n)$$

So since:

$$\sum_{c \in \mathcal{C}} |\mathcal{S}^\dagger(c) \setminus \{r\}| \leq T$$

we have, by the union bound, that with probability at least $1 - 2T \exp(-2\hat{\epsilon}^2 n)$, which is no less than $1 - \delta$, we have:

$$|\delta'(c, x)| \leq \hat{\epsilon}$$

for all $c \in \mathcal{C}$ and $x \in \mathcal{S}^\dagger(c)$. So assume that this is the case. Now take any policy $\pi \in \mathcal{Q}$ which has statistical parity with respect to $\rho^*$. For all $c \in \mathcal{C}$ define:

$$\zeta(c) := \sum_{v \in \mathcal{V}} \rho^*(c, v)$$

For all $(c, a) \in \mathcal{C} \times \mathcal{A}$ and $\mathcal{Y} \in \mathcal{S}^\dagger(c)$ define:

$$q^*(c, \mathcal{Y}, a) := \frac{1}{\mu^*(c, \mathcal{Y})\zeta(c)} \sum_{v \in \mathcal{Y}} \rho^*(c, v)\pi(c, v, a)$$

$$q(c, \mathcal{Y}, a) := \frac{1}{\tilde{\mu}(c, \mathcal{Y})\zeta(c)} \sum_{v \in \mathcal{Y}} \rho(c, v)\pi(c, v, a)$$

$$\beta(c, \mathcal{Y}, a) := q^*(c, \mathcal{Y}, a) - q(c, \mathcal{Y}, a)$$

and let $h(c, \mathcal{Y})$ be the height of $\mathcal{Y}$ in the tree $\mathcal{S}^\dagger(c)$. Note that by definition of $\mu^*(c, \mathcal{Y})$ we have:

$$q^*(c, \mathcal{Y}, a) \leq \frac{1}{\mu^*(c, \mathcal{Y})\zeta(c)} \sum_{v \in \mathcal{Y}} \rho^*(c, v) = 1$$

We take the inductive hypothesis that for all $x \in \mathcal{S}^\dagger(c)$ we have:

$$|\beta(c, x, a)| \leq 2h(c, x)\hat{\epsilon}$$

and prove by induction on $h(c, x)$. The inductive immediately holds for $h(c, x) = 0$ by definition of $\rho$. Now suppose it holds when $h(c, x) = h'$ (for some $h' \in [H]$) and consider $x$ with $h(c, x) = h' + 1$. Note that:

$$q^*(c, x, a) = \frac{1}{\mu^*(c, x)} \left(\mu^*(c, \triangleleft(x))q^*(c, \triangleleft(x), a) + \mu^*(c, \triangleright(x))q^*(c, \triangleright(x), a)\right)$$
$$= p^*(c, \triangleleft(x))q^*(c, \triangleleft(x), a) + p^*(c, \triangleright(x))q^*(c, \triangleright(x), a)$$

and that:

$$q(c, x, a) = \frac{1}{\tilde{\mu}(c, x)} \left(\tilde{\mu}(c, \triangleleft(x))q(c, \triangleleft(x), a) + \tilde{\mu}(c, \triangleright(x))q(c, \triangleright(x), a)\right)$$
$$= p(c, \triangleleft(x))q(c, \triangleleft(x), a) + p(c, \triangleright(x))q(c, \triangleright(x), a)$$
$$= p(c, \triangleleft(x))(q^*(c, \triangleleft(x), a) - \beta(c, \triangleleft(x), a)) + p(c, \triangleright(x))(q^*(c, \triangleright(x), a) - \beta(c, \triangleright(x), a)))$$

so, by defining:

$$\hat{\beta}(c, x, a) := p(c, \triangleleft(x))\beta(c, \triangleleft(x), a) + p(c, \triangleright(x))\beta(c, \triangleright(x), a)$$

we have:

$$\beta(c, x, a) := q^*(c, x, a) - q(c, x, a)$$
$$= (p^*(c, \triangleleft(x)) - p(c, \triangleleft(x)))q^*(c, \triangleleft(x), a) + (p^*(c, \triangleright(x)) - p(c, \triangleright(x)))q^*(c, \triangleright(x), a) + \hat{\beta}(c, x, a)$$
$$= \delta'(c, \triangleleft(x))q^*(c, \triangleleft(x), a) + \delta'(c, \triangleright(x))q^*(c, \triangleright(x), a) + \hat{\beta}(c, x, a)$$

so that, by the inductive hypothesis and since $h(c, \triangleleft(x)) = h(c, \triangleright(x)) = h'$, we have:

$$|\beta(c, x, a)| \leq |\delta'(c, \triangleleft(x))|q^*(c, \triangleleft(x), a) + |\delta'(c, \triangleright(x))|q^*(c, \triangleright(x), a) + |\hat{\beta}(c, x, a)|$$
$$\leq \hat{\epsilon}q^*(c, \triangleleft(x), a) + \hat{\epsilon}q^*(c, \triangleright(x), a) + |\hat{\beta}(c, x, a)|$$
$$\leq 2\hat{\epsilon} + |\hat{\beta}(c, x, a)|$$
$$\leq 2\hat{\epsilon} + p(c, \triangleleft(x))|\beta(c, \triangleleft(x), a)| + p(c, \triangleright(x))|\beta(c, \triangleright(x), a)|$$
$$\leq 2\hat{\epsilon} + 2p(c, \triangleleft(x))h'\hat{\epsilon} + 2p(c, \triangleright(x))h'\hat{\epsilon}$$
$$= 2(1 + h')\hat{\epsilon}$$
$$= 2h(c, x)\hat{\epsilon}$$

We have hence proved that the inductive hypothesis holds always. In particular it holds for $x = r$ so that:

$$|\beta(c, r, a)| \leq 2H\hat{\epsilon} = \epsilon/2 \tag{21}$$

Since $\mu^*(c, r) = \tilde{\mu}(c, r) = 1$ and $r = \mathcal{V}$ this translates to:

$$\left|\frac{1}{\zeta(c)} \sum_{v \in \mathcal{Y}} \rho^*(c, v)\pi(c, v, a) - \frac{1}{\zeta(c)} \sum_{v \in \mathcal{Y}} \rho(c, v)\pi(c, v, a)\right| \leq \epsilon/2$$

Since $\pi$ has statistical parity with respect to $\rho$ there exists some function $\kappa : \mathcal{A} \to [0, 1]$ such that:

$$\frac{1}{\zeta(c)} \sum_{v \in \mathcal{Y}} \rho(c, v)\pi(c, v, a) = \kappa(a) \quad \forall (c, a) \in \mathcal{C} \times \mathcal{A}$$

which means:

$$\kappa(a) - \epsilon/2 \leq \frac{1}{\zeta(c)} \sum_{v \in \mathcal{Y}} \rho^*(c, v)\pi(c, v, a) \leq \kappa(a) + \epsilon/2$$

for all $c \in \mathcal{C}$. Hence, $\pi$ has $\epsilon$-approximate statistical parity with respect to $\rho^*$. This completes the proof that $\rho$ is $\epsilon$-close to $\rho^*$.

We now show that each policy $\pi_t^*$ has statistical parity with respect to $\rho$. Take any $c \in \mathcal{C}$ and let $\mathcal{L}_t(c)$ be the set $\mathcal{L}(c)$ at trial $t$. Note that for all $\mathcal{Y} \in \mathcal{L}^\dagger(c)$ we have:

$$\tilde{\mu}(c, \mathcal{Y}) = \frac{1}{\zeta(c)} \sum_{v \in \mathcal{Y}} \rho(v)$$

so since for all $\mathcal{Y} \in \mathcal{S}^\dagger(c) \setminus \mathcal{L}^\dagger(c)$ we have:

$$\tilde{\mu}(c, \mathcal{Y}) = \tilde{\mu}(c, \triangleleft(\mathcal{Y})) + \tilde{\mu}(c, \triangleright(\mathcal{Y}))$$

we have by induction that for all $\mathcal{Y} \in \mathcal{S}^\dagger(c)$:

$$\tilde{\mu}(c, \mathcal{Y}) = \frac{1}{\zeta(c)} \sum_{v \in \mathcal{Y}} \rho(v)$$

This means that for all $a \in \mathcal{A}$ we have:

$$\frac{1}{\zeta(c)} \sum_{v \in \mathcal{V}} \rho(v)\pi_t^*(c, v, a) = \frac{1}{\zeta(c)} \sum_{v \in \mathcal{V}} \rho(v)\pi_t(c, \chi_t(c, v), a)$$

$$= \frac{1}{\zeta(c)} \sum_{\mathcal{Y} \in \mathcal{L}_t(c)} \sum_{v \in \mathcal{Y}} \rho(v)\pi_t(c, \chi_t(c, v), a)$$

$$= \frac{1}{\zeta(c)} \sum_{\mathcal{Y} \in \mathcal{L}_t(c)} \sum_{v \in \mathcal{Y}} \rho(v)\pi_t(c, \mathcal{Y}, a)$$

$$= \sum_{\mathcal{Y} \in \mathcal{L}_t(c)} \tilde{\mu}(c, \mathcal{Y})\pi_t(c, \mathcal{Y}, a)$$

$$= \sum_{x \in \mathcal{G}_t(c)} \mu_t(c, x)\pi_t(c, x, a)$$

so since we have $\pi_t \in \mathcal{F}(\mu_t)$ we have that $\pi_t^*$ has statistical parity with respect to $\rho$.

We now prove the regret bound. Take any $\tilde{\pi} \in \mathcal{Q}$ with statistical parity with respect $\rho$. For all $c \in \mathcal{C}$ let $\mathcal{L}'(c)$ be the minimal cardinality permitted clustering such that for all sets $\mathcal{Y} \in \mathcal{L}'(c)$ and all $u, v \in \mathcal{Y}$ we have $\tilde{\pi}(c, u, \circ) = \tilde{\pi}(c, v, \circ)$. Now we define $\tilde{\pi}' \in \mathcal{P}$ as follows. First extend $\tilde{\mu}$ such that for all $(c, \mathcal{Y}) \in \mathcal{C} \times \mathcal{X}$ we have:

$$\tilde{\mu}(c, \mathcal{Y}) = \frac{1}{\zeta(c)} \sum_{v \in \mathcal{Y}} \rho(v)$$

For any descendant $\mathcal{Z}$ of some $\mathcal{Y} \in \mathcal{L}'(c)$ we define $\tilde{\pi}'(c, \mathcal{Z}, \circ) = \tilde{\pi}(c, u, \circ)$ for any $u \in \mathcal{Y}$. For any $x$ that is a proper ancestor of some element of $\mathcal{L}'(c)$ we define $\tilde{\pi}'(c, x, \circ)$ so that:

$$\tilde{\pi}'(c, x, a) := \frac{\tilde{\mu}(c, \triangleleft(x))\tilde{\pi}'(c, \triangleleft(x), a) + \tilde{\mu}(c, \triangleright(x))\tilde{\pi}'(c, \triangleright(x), a)}{\tilde{\mu}(c, x)}$$

Note that by induction on the height of $x$ we have:

$$\sum_{a \in \mathcal{A}} \tilde{\pi}'(c, x, a) = \frac{\tilde{\mu}(c, \triangleleft(x)) + \tilde{\mu}(c, \triangleright(x))}{\tilde{\mu}(c, x)} = 1$$

so that $\tilde{\pi}' \in \mathcal{P}$ as required. Given $x \in \mathcal{T}$ define $\Downarrow(x)$ to be the leaf descendants of $x$. By induction on the height of $x$ in $\mathcal{T}$ we have:

$$\tilde{\pi}'(c, x, a) = \frac{1}{\tilde{\mu}(c, x)} \sum_{x' \in \Downarrow(x)} \tilde{\mu}(c, x')\tilde{\pi}'(c, x', a)$$

so that, since the leaves are the singleton sets, we have, for all $\mathcal{Y} \in \mathcal{X}$, that:

$$\tilde{\pi}'(c, \mathcal{Y}, a) = \frac{1}{\tilde{\mu}(c, \mathcal{Y})\zeta(c)} \sum_{v \in \mathcal{Y}} \rho(c, v)\tilde{\pi}(c, v, a)$$

Hence, on trial $t$ we have:

$$\sum_{x \in \mathcal{G}_t(c)} \mu_t(c, x)\tilde{\pi}'(c, x, a) = \frac{1}{\zeta(c)} \sum_{\mathcal{Y} \in \mathcal{L}_t(c)} \sum_{v \in \mathcal{Y}} \rho(c, v)\tilde{\pi}(c, v, a) = \frac{1}{\zeta(c)} \sum_{v \in \mathcal{V}} \rho(c, v)\tilde{\pi}(c, v, a)$$

so since $\tilde{\pi}$ has statistical parity with respect to $\rho$ we have that $\tilde{\pi}' \in \mathcal{F}(\mu_t)$.

Now that we have shown that $\rho \in \bigcap_{t \in [T]} \mathcal{F}(\mu_t)$ we can apply the regret bound in Theorem 4.1. Note that for all $c \in \mathcal{C}$ the number of edges $(x, x')$ in $\mathcal{T}$ in which $\tilde{\pi}'(c, x, \circ) \neq \tilde{\pi}'(c, x', \circ)$ is at most $2\psi(c, \tilde{\pi}) - 2$. Hence, as in Section D.1 (noting that $\log(|\mathcal{T}|) \leq H$), we have a distribution $\vartheta^* : \mathcal{C} \times \mathcal{H} \to [0, 1]$ with:

$$\tilde{\pi}'(c, x, a) = \sum_{e \in \mathcal{H}} [\![e(x) = a]\!] \vartheta^*(c, e) \quad \forall (c, x, a) \in \mathcal{C} \times \mathcal{X} \times \mathcal{A}$$

and:

$$\sum_{c \in \mathcal{C}} \sum_{e \in \mathcal{H}} \vartheta^*(c, e) \ln\left(\frac{\vartheta^*(c, e)}{\vartheta(e)}\right) \in \mathcal{O}(H \ln(K)\Psi(\tilde{\pi}))$$

Noting also that we have at most $2n\Psi(\tilde{\pi})$ trials where $\tilde{\pi}'(c_t, x_t, \circ) \neq \tilde{\pi}(c_t, v_t, \circ)$ we obtain, from Theorem 4.1, the regret bound.

