# OpenReview forum: "Statistical Parity with Exponential Weights"
_NeurIPS.cc/2025/Conference — NeurIPS 2025 poster_

### Official Review · Reviewer_t2Fz · 2025-07-03

**Clarity:** 3
**Significance:** 3
**Originality:** 3
**Rating:** 5
**Confidence:** 3

**Summary:**

The main contribution of this paper is developing a meta-algorithm called SPEW that translates an implementation of the HEDGE/EXP4 into a scheme that (i) ensures statistical parity, and (ii) achieves the same regret as EXP4 against the best parity-achieving policy in hindsight. The authors instantiate this general strategy for various examples, such as for graph valued contexts, finite metric spaces, statistical parity w.r.t. the empirical distribution, and for approximate statistical parity for hierarchical clustering.
Finally, the authors conclude by discussing how SPEW can be applied to a fair classification problem via a online-to-batch conversion.

**Questions:**

**Q1:** Can you discuss about the possibility of constructing SPEW-like meta-algorithms for other notions of fairness, such as equalized odds for example.

**Q2:** In Line 68 it is mentioned that the underlying methodology of SPEW can be generalized beyond HEDGE-based implementations. Can you expand upon this a bit.

**Q3:** Since both the time and space complexity of SPEW in each trial grows linearly with $M$, can you discuss potential ways of dealing with continuous or finite-but-large class of protected characteristics? What additional assumptions might we need to make?

---
### Minor Questions/Comments

Line 23: The notation used in the display here is a bit confusing: if I understand correctly, a, c, v are random variables, and b, d, d' refer to realizations?

Lines 446, 659, 740: Broken cross-references.

**Ethical Concerns:**

["NO or VERY MINOR ethics concerns only"]

**Final Justification:**

I enjoyed reading this paper, as well as following the discussion with other reviewers and the authors. The authors have clarified my concerns, and I will keep my score and recommend accept.

**Limitations:**

Yes.

**Paper Formatting Concerns:**

None.

**Quality:**

3

**Strengths And Weaknesses:**

Strengths

* I think the main strength of the paper is that proposed idea of a wrapper around HEDGE algorithm is conceptually quite simple and elegant. Yet, this scheme seems to be applicable quite broadly to structured and unstructured problems, with known or unknown distribution $\rho$.

Weaknesses

* One minor issue could be the fact that the reduction is tied to a very specific notion of fairness (i.e., statistical parity), and it doesn't seem obvious to extend it to other notions of fairness.

* The paper only consists of theoretical results, and there is no empirical evaluation of practical issues like the effect of constant factors, the speed of convergence, any trade-offs between fairness and performance, scalability etc.

---

> ### Author Rebuttal · Authors · 2025-07-30
>
> We thank you for your review - here are our responses.
>
>
> > One minor issue could be the fact that the reduction…
>
> We believe that the main idea behind SPEW -  a “policy processing” step (which takes the output of Hedge for each virtual-context/protected-characteristic pair and maps it to a constrained policy) followed by taking an unbiased estimate of the gradient of a surrogate convex loss, splitting it up into effective loss vectors, and feeding it back into Hedge, should be applicable to other fairness notions. In particular, we believe (but are not 100 percent sure that) it should be applicable to the constraint of selecting each action with a specific (or bounded) overall probability, which is crucial for applications such as inclusive hiring for jobs. However, we do believe that to get regret bounds here the context selection by Nature must be i.i.d. stochastic (as opposed to the fully adversarial Nature of SPEW).
>
>
> > The paper only consists of theoretical results…
>
> We felt that since this is, to the best of our knowledge, the first paper to enforce statistical parity in the bandit setting, there is no basis for comparison in experiments. We do note that in the appendix we analyse the constant factor in the regret and it is very small - only $\sqrt{8}$ when the learning rate is optimally tuned. As for trade-offs between fairness and performance, this paper is focused on achieving statistical parity exactly (if the parity distribution is known). We leave it to follow-up work to widen the constraint set allowing for approximate statistical parity with a regret bound against any comparator in the widened set.
>
>
> > Q1 Can you discuss the possibility…
>
> Please see above where we give an example of another fairness application that we believe the underlying concepts can be applied to. Let us think more about other notions of fairness and we’ll get back to you in the discussion phase.
>
>
> > In Line 68…
>
> We would like to clarify that this statement (or rather, the existence of theoretical bounds) is conjecture, which is why we use the word “should”. An intuition is that SPEW works by taking the outputs of Hedge and performing a “policy processing” step, and then using the (unbiased estimator of the) derivative of the surrogate convex function in mirror descent. One should be able to replace Hedge (i.e. mirror descent on the simplex) by an algorithm that still performs something similar to mirror descent (with (unnormalised) KL divergence) but not over the simplex - such as the CBA algorithm of [18]. For CBA we have a set of predictors that each assign a confidence-rated prediction to each context and the regret bound would be with respect to any linear combination of predictors that satisfies statistical parity (the existence of such linear combinations can be enforced by adding specialists that are awake only on a particular context).
>
> We also believe that, when we have full information feedback, SPEW could be applied to a contextualised version of the recent ReSeT algorithm of “Online Convex Optimisation: the Optimal Switching Regret for every Segmentation Simultaneously”. This would allow us to adapt to non-stationarity whilst having inductive biases and will also sharpen the regret when there is no inductive bias. The application should be achieved by choosing the base (static) algorithm for ReSeT to be SPEW. By constructing the propagating actions of ReSeT one will end up with a convex combination of policies with statistical parity - which is itself a policy with statistical parity.
>
> We will think through the details of the proofs of CBA and ReSeT to check that there is nothing that gets in the way and get back to you in the discussion phase.
>
> However, there is an exponential weight algorithm in which it is certainly not straightforward (and may well be impossible) to analyse the application of the SPEW methodology to - and that is the specialist algorithm of “Using and Combining Predictors that Specialise”. It may be possible to get theoretical results here but there is an issue in that we must update all specialists on each trial rather than just the awake ones (which causes issues as we are no longer using mirror descent). We shall discuss specialists as an open problem in the revised version.
>
>
> > Since both the time and space complexity…
>
> Having continuous (or an extremely large number of) protected characteristics is a hard one. All we can think of at this point is to discretise the protected characteristics. However, this would only give you approximate statistical parity and the approximation would depend on how smoothly the marginal parity distribution changed with the protected characteristic. However, we will think more on this problem and if we think of anything else we will let you know in the discussion period.
>
>
> > Line 23: The notation used in…
>
> Yes exactly. Thanks for giving us a description to add to the paper.
>
>
> > Broken cross-references.
>
> Thank you - we will fix these.

---

> > ### Comment · Reviewer_t2Fz · 2025-08-05
> >
> > Thank you for your detailed response that helped me understand the work better. I think it would be nice if you include the above paragraph about generalizing SPEW to other notions of fairness (even as an informal conjecture) in the final version.

---

> > > ### Author Response · Authors · 2025-08-06
> > >
> > > Thank you - we shall certainly include this paragraph.

---

> > > > ### Author Response · Authors · 2025-08-09
> > > >
> > > > We thought further about CBA and ReSeT and saw no obstacles to the application of the SPEW methodology. However, we cannot be totally sure without writing down the proofs.

---

### Official Review · Reviewer_Fh2J · 2025-07-03

**Clarity:** 1
**Significance:** 3
**Originality:** 2
**Rating:** 4
**Confidence:** 3

**Summary:**

The paper develops a novel algorithm for the contextual bandit setting, which is capable of producing no-regret policies that achieve guaranteed statistical parity with respect to protected attributes. The algorithmic proposal starts with Hedge (expressed in oracle-call format) and via reductions/tricks specific to the setting of obtaining statistical parity builds an algorithm entitled SPEW on top of it. The algorithm is then shown to provide regret bounds that have mild dependence on the dimensionality of the problem (rather than blow up when said dimension is exponential).

The algorithm's dimensionality properties are then showcased in several structured settings including: graphs with homophily; hierarchical settings; metric-induced settings. By way of straightforward extensions, it is furthermore observed that the proposed algorithm also gives a novel solution to the same problem even in the batch (offline i.i.d.) setting, as well as that an empirical-distribution variant of SPEW can be constructed that doesn't require full access to the underlying feature distribution (only to the observed samples).

**Questions:**

Please see the Weaknesses section above for a list of concrete items that in my view would significantly improve readability of the paper.

Specifically, the following points are of primary importance:

--- A more careful discussion of related work such as on long-term constraints intersect fairness in online learning; in particular emphasizing technical details compared to methods achieving cumulative constraint violation regret bounds with different tradeoffs in the long-term literature.

--- A clearer reorganized structure where SPEW comes first, before the examples; and please consider throwing both online to batch and also the empirical version of the algorithm into a dedicated section called Extensions, to separate from Examples.

--- SPEW should be re-discussed with a substantial degree of insight into why the particular elements of the pseudocode were chosen as they are (included but not limited to the \lambda); for this one, please keep in mind first-time readers!

I expect that the authors will be able to provide a careful list of patches along those lines, as well as concrete and detailed chunks of text to be later included in the revised manuscript, in the rebuttal. If this is not performed, then, for full transparency, I may downgrade my current (advance) borderline-accept to borderline-reject.

**Ethical Concerns:**

["NO or VERY MINOR ethics concerns only"]

**Final Justification:**

I have carefully considered the authors' response to my review. I have also studied the other reviews and discussion. My overall assessment is that once the presentational aspects of this paper are improved, it will become an acceptable part of the conference. The contribution to the field of online fairness is there, though it fits into a substantial line of work on other notions of fairness and on long-term constraints, so there is no overwhelming technical novelty that would sway me towards a higher score. These aspects form the basis of my borderline assessment in this case. (The other reviews, in my opinion, are somewhat too optimistic in terms of assigned scores to the paper, given its current less-readable state and relatively streamlined techniques; however, the contribution seems appropriate for NeurIPS and valid so far as I checked, so I would not wish to argue for rejection.)

**Limitations:**

Yes.

**Quality:**

2

**Strengths And Weaknesses:**

The main strength of the method is that it fills a niche that to my knowledge has not yet been addressed in the fairness-in-bandits literature; in particular,

--- It enforces fundamental statistical parity guarantees, which have not been accomplished for this task even in the i.i.d. offline scenario;

--- Said statistical parity guarantees are obtained and satisfied in an exact way, without the corresponding O(\sqrt{T}) or similar regret term corresponding to constraint violation;

--- In contrast to some other related strands of research such as many of the long-term constraints papers, the present work operates under bandit rather than e.g. full information feedback regime;

--- The provided worked-out examples (and the general guarantees) show that the algorithm successfully deals with a-priori exponentially-high-dimensional settings by being able to rely on certain kinds of structure in them (while this is a pervasive theme throughout many online bandits literatures, it has not been instantiated in such setting before, to my knowledge).

For the main weakness, I have to point out the composition and writing of this paper; it is far from clearly or lucidly written, and has required a lot of parsing effort that could have been saved with a different organization/writing. To give some concrete pointers:

--- First, the main algorithm (SPEW) is relegated to the depths of the paper (after an entire array of example constructions), which I couldn't get behind.

--- Second, and very importantly, the algorithm SPEW that's the main contribution of the paper is not described insightfully in the main paper, nor even in the appendix! For instance, a vague promise is made that the meaning of the concretely set variables \lambda can be understood from the algorithm analysis, but even in the corresponding appendix this is explained in a very murky way! This opacity could have absolutely been avoided --- e.g. along the lines of explaining that we are looking for suitably upper- and lower-bounding surrogate losses, which, in turn, leads to a loss of a particular form whose subgradient is of the form... etc etc.

--- Third, the comparisons to prior literature are brief and incomplete in ways that make them rushed/not as insightful as they should be in this set of circumstances. To be sure, some effort is made in the intro --- by asking the readers to imagine the process of forming the Lagrangian for the corresponding parity-constrained problem and solving it for the primal and dual variables, and pointing out that this is inefficient to do in the "brute-force" way. Such a point is best understood by someone with a background in long-term constraints; but then for such readers, the following couple sentences hinting at the differences from existing long-term works (that most of them don't achieve exact constraint satisfaction and many of them are not applicable to the bandit setting) are not discussed deeply enough to explain the technical nuances. (In fact, some long-term constraints setups emphasize (near-)exact constraint satisfaction in general settings, and some other such works do target bandit settings; and yet other long-term constraints papers have some degree of dimensionality-avoiding oracle efficiency to them.) Also, some essential fairness in online learning references have been missed, such as works on individual fairness online (like Bechavod et al, Metric-free individual fairness).  Moreover, down the line this leads to the proofs of the algorithm's analysis in which long-term constraints influences are not invoked when appropriate. E.g. surrogate loss constructing techniques, which this paper's analysis has an instance of, could/should be mentioned at the right point.

---

> ### Author Rebuttal · Authors · 2025-07-30
>
> We thank you for your detailed review and helpful suggestions. We are committed to improving the paper along the lines you’ve proposed. You mentioned that you would like to see specific chunks of text for the revised version, but we weren’t sure which parts you had in mind. If you could clarify what additional text you’d like to see, we would be happy to draft and share it during the discussion period. We have, however, included one chunk of important text.
>
> We believe that, with your suggestions in mind, the paper can be presented in a much clearer and more effective way. We are fully committed to revising the presentation accordingly and would be grateful for any additional guidance on what you would like to see in the revised version.
>
>
> > First, the main algorithm (SPEW) is relegated to the depths of the paper.
>
> We chose to showcase the results of the paper early in the main body, which is why the examples appear first and the details of how and why SPEW works are deferred to the appendix. That said, we agree that this is not optimal. We propose the following modifications. We will cut the pseudocode section from the main body and relegate it to the appendix. With the addition of the extra page we will write a detailed description of how and why the algorithm works. We will place this description before the examples section. The description will contain the full algorithm with all the equations involved.
>
>
> > Secondly, and very importantly…
>
> Please see the above for details of how we will introduce an accessible description of the algorithm into the main body. We propose to start the description with this (draft) chunk of text…
>
> “Before presenting the technical details, we begin with a high-level overview of the main steps in SPEW.  SPEW operates with any implementation of Hedge that satisfies the inductive bias. A separate instance of this Hedge implementation is maintained for each protected characteristic.
>
> At the start of trial $t$ each virtual context is revealed in turn to each Hedge instance, which gives us a distribution over the actions for each virtual-context/protected-characteristic pair. This forms a (virtual) policy that we refer to as the “raw” policy. In general, however, the raw policy will not satisfy statistical parity.
>
> The final policy $\pi_t$ is obtained by applying a procedure we call “policy processing” to the raw policy (details to follow). We will define a convex function $y_t$, of the raw policy, which upper bounds the expected loss of $\pi_t$ (evaluated at the context $x_t$). Importantly, for any policy that satisfies statistical parity, $y_t$ evaluated at that policy lower bounds its expected loss (evaluated at the context $x_t$). These properties allow us to use $y_t$ as a surrogate for the expected final loss incurred by the raw policy.
>
> The update step proceeds by computing an unbiased estimator of the gradient of $y_t$ evaluated at the raw policy. This unbiased estimator gives us, for each virtual-context/protected-characteristic pair, a vector over the actions (which replaces the loss vector in a Hedge update). Hence, for each protected characteristic, the corresponding instance of Hedge is updated once for each virtual context.”
>
> After this chunk of text we shall go into the details of the policy processing step, followed by the definition of $y_t$ and the unbiased estimator of its gradient. We hope to have enough room to write all the lemmas in Appendix B as equations in order to give the reader a feel of how the result is proved. We will make sure that we make the description as accessible to readers as possible.
>
>
> > Third, the comparisons to priori literature…
>
> We commit ourselves to doing a far more comprehensive review of the related literature, which will be placed in the appendix.
>
> When it comes to the literature on long-term constraints we do note that we briefly described the results of the papers [4] and [23] which, to the best of our knowledge, are state of the art. The paper [4] does indeed use an oracle to handle the high dimensional bandit problem efficiently. However, the statistical parity is violated and the regret is far worse than SPEW. We note that for [23] the regret is $O(\sqrt{T})$ and not $O(1)$ as the constraint set has empty interior (which we will explain in the paper). [23] is actually irrelevant as it is not designed for bandit or high dimensional problems. We will make a dedicated section in the appendix that will give a comprehensive discussion of long-term constraint algorithms and how they work, highlighting the differences between these works and ours. We will make sure we cite many works in this appendix section. We will also mention any ties between our work and long-term constraint works, like the use of a surrogate loss as you mentioned.
>
> We will give a full review of fairness in the batch setting. We will also include discussions on individual fairness and we thank you for the citation here.
>
> If there are specific papers that you believe should be cited we would be very grateful for your suggestions.
>
>
> > Specifically, the following points are of primary importance...
>
> Please see above. We shall be implementing all your suggestions fully.
>
>
> > and please consider throwing both online-to-batch...
>
> Thank you for the suggestion. Our intention was to present the two unknown-distribution algorithms (the "empirical statistical parity" and the hierarchical decomposition) as direct instances of SPEW, which is why we placed them in the examples section. That said, we’re open to restructuring if you feel that it would improve clarity.

---

> > ### Comment · Reviewer_Fh2J · 2025-08-06
> >
> > Thank you for providing this response, I appreciate your commitment to improving various presentation aspects of the paper. As you can also see from the other reviewers' points, there seems to be a consensus that the paper needs to be made more accessible in various substantial ways.
> >
> > At the same time, I trust that you will be able to successfully perform these changes: The excerpt of a new algorithm overview that you provided sounds excellent to me, and the paper should generally acquire and maintain that level of clarity throughout.
> >
> > As regards the examples section and including the online-to-batch construction in it, this is a somewhat minor concern compared to a more transparent overview of the algorithm itself and the proof techniques. That said, I do think it would be good to separate the genuine examples from online-to-batch --- as the latter is a standard technicality in online learning and thus does not exemplify specific properties of your algorithm nor does it add anything interesting to the fairness discussion (I do understand that you wish to highlight that this was not available even in the batch setting prior to this work, but a remark would suffice for the main part).
> >
> > Finally, as regards related works to consider, on the long-term constraints literature front I would start with Mahdavi et al (2012, JMLR) which is the originator of that line of work, followed by works such as Jenatton et al (2016), and followed by more modern developments such as Cao and Liu (I don't suggest doing a comprehensive discussion of all these works obviously, but documenting the original development of techniques/bounds/proofs is important for the reader's background on where this work fits in; currently you only give a couple very recent references such as Castiglioni et al and Yu and Neely). On the fairness-in-online-learning side, there's more references to consider, the one mentioned was just one important example. E.g. followups on that paper are a good place to start exploring for a better overview of the field of distinct fairness notions online; see other works like Xu et al (2023, AISTATS), Doubly Fair Dynamic Pricing; Individual Fairness in Hindsight (2021, JMLR), and others --- there are various connections and differences in each case that I'd urge you to elicit for the reader's benefit.
> >
> > With these and other modifications for readability, the manuscript would improve substantially, so I maintain my score.

---

> > > ### Author Response · Authors · 2025-08-06
> > >
> > > Thank you very much for your helpful advice.

---

### Official Review · Reviewer_FevA · 2025-07-03

**Clarity:** 3
**Significance:** 3
**Originality:** 4
**Rating:** 5
**Confidence:** 2

**Summary:**

In this paper, the authors propose a meta-algorithm that can transform any efficient implementation of Hedge (or discrete Bayesian inference algorithm) into a contextual bandit algorithm that guarantees exact statistical parity on every trial.

**Questions:**

I am not familiar with the theory things about contextual bandits, so I only have some questions from the angle of fairness.
1. As statistical fairness focus on the "positive" action, in the presented examples in Sec 4, can a "positive" action always exist and be clearly defined, especially when |A|>2?
2. In the second example in Introduction, it says "the choice of blue agent depends on certain private attributes of the system" and "enforcing statistical parity ensures that the selected blue agent is independent of the sensitive attribute". In this sense, the correlation between the blue agent and private attributes is broken, which may hurt the model's performance (if any), as some useful features can no longer be utilized. How does the proposed method address this potential trade-off?

**Ethical Concerns:**

["NO or VERY MINOR ethics concerns only"]

**Final Justification:**

I am not very confident about the theoretical correctness. But this paper looks good and no reviewer is negative so far. I think this paper is ready to be accepted.

**Limitations:**

yes

**Quality:**

3

**Strengths And Weaknesses:**

***I have reviewed the paper with care, but evaluating the correctness of the theoretical analysis is beyond my current expertise. ***

---

> ### Author Rebuttal · Authors · 2025-07-30
>
> We thank you for your review - here are our responses.
>
>
> > As statistical parity focuses on the positive action
>
> The formulation of statistical parity in terms of a “positive action” typically applies in settings with two actions so that $K=2$. In fact, for statistical parity there is nothing special about the positive action - if we flipped positive and negative we still have statistical parity. Our framework generalises the notion to any $K$ - the selected action is independent of the protected characteristic. There is another generalisation though - we could keep a positive action and the definition is that the probability that the positive action is selected is equal across all protected characteristics (which is a weaker constraint). We believe that it is this generalisation that you have in mind. We also believe that it should be straightforward to modify SPEW so that it satisfies this constraint and has the regret bound against any policy satisfying the constraint.
>
>
> > In the second example…
>
> The constraint here is that we must select the blue agent in such a way that no information about the sensitive attribute is leaked. Since the blue agent must satisfy this constraint, performance will indeed be less than if it was allowed to leak information. However, aside from the regret term for learning, our algorithm randomly chooses the blue agents that perform (in expectation) the best given this constraint. We do indeed use all the features in order to do this. Interestingly, simply ignoring the sensitive attribute does not satisfy the constraint - as information can still leak due to the correlation of the sensitive attribute and the other attributes.

---

> > ### Comment · Reviewer_FevA · 2025-08-07
> >
> > Thank you for addressing my concerns. This paper is highly theoretical, and many parts are beyond my ability to evaluate. I have kept a low confidence score to allow the AC to better summarize the final decision.

---

### Official Review · Reviewer_rm31 · 2025-07-18

**Clarity:** 2
**Significance:** 3
**Originality:** 3
**Rating:** 4
**Confidence:** 2

**Summary:**

In this paper, the authors propose an algorithm for contextual bandits that enjoys statistical parity. The proposed algorithm uses a Hedge algorithm subroutine for each protected attribute.

**Questions:**

No specific questions from my side, but I would appreciate an accessible (to say someone familiar with a textbook treatment of Hedge/Exp4) and intuitive explanation on how SPEW achieves statistical parity in the present context.

**Ethical Concerns:**

["NO or VERY MINOR ethics concerns only"]

**Limitations:**

Yes

**Quality:**

3

**Strengths And Weaknesses:**

The paper makes an interesting contribution, IMO. However, I think this articulation is too dense for a non-expert. I had a hard time trying to understand the basic idea of how the statistical parity is achieved by the algorithms.

Even the examples in Section 3 are not covered in sufficient detail for a reader who has not encountered these problems before.

Overall, I think this paper is an instance of there being too much material being packed into too little space.

Minor suggestion: In the paragraph starting on Line 40 from the introduction, the authors describe the contextual bandit problem. It would help if the authors could also articulate what statistical parity means for this specific model (this is made clear later though).

---

> ### Author Rebuttal · Authors · 2025-07-30
>
> We thank you for your review - here are our responses.
>
>
> > I had a hard time trying to understand the basic idea of how statistical parity is achieved by the algorithms.
>
> In Appendix B we give a detailed overview of how and why the meta-algorithm SPEW works, although we admit this may not be as accessible as possible. In order to make things more accessible (and in the main body) we shall relegate the pseudocode (Section 5.3) to the appendix and replace it by a complete description of how and why SPEW works (using some of the extra page if necessary). This description will be made as accessible to the reader as possible and will complement Appendix B, which contains all the intermediate results.
>
>
> > Even the examples in Section 3 are not covered in sufficient detail…
>
> We agree - this was due to space issues. We will use the extra page to expand section 3 in order to give more details and make things more accessible to the reader. Both graphs and metric spaces work via reduction to a (potentially random) tree (which is, to the best of our knowledge, the state of the art approach for adversarial bandits in graphs and metric spaces). Hence, in order to save space, we propose to unify sections 4.1 and 4.2 by giving the results for a tree in the main body, then briefly discuss the reductions to it, and defer the actual results for graphs and metric spaces to the appendix. Since the tree is a simpler object to describe, this will make things more accessible to the reader in the main body. In Section 4.4 we hope to have enough space to describe how Euclidean space can be hierarchically clustered (the clusters are hyper-cuboids and you keep splitting evenly across the longest axis)
>
>
> > Overall, I think this paper is an instance of there being too much material…
>
> Yes we totally agree. We hope that the extra page and our intended changes described above will give the paper more room to breathe.
>
>
> > Minor suggestion: In the paragraph starting on Line 40…
>
> Thanks for the suggestion - we will incorporate.
>
>
> > No specific questions from my side, but I would appreciate…
>
> As described above, we will do this by relegating the pseudocode to the appendix.

---

### Note · Authors · 2025-08-15

We would like to thank the reviewers for their appreciation of our paper and their thoughtful and constructive comments. We are committed to improving the presentation of the paper, and will include a full and accessible description of how and why SPEW works in the main body (relegating the pseudocode to the appendix to make space). We will make sure that this description comes before the examples section. We will also greatly expand the literature review, which will give a much more comprehensive coverage of works on long-term constraints and fairness in online learning.

---

### Decision · Program_Chairs · 2025-09-17

**Decision:**

Accept (poster)

**Comment:**

The paper studies contextual bandits, where part of the context encodes a sensitive attribute. They provide regret bounds that satisfy statistical parity in this setting.

+ The algorithm and setting is different from existing work in group fairness in bandits, and the results seem better i.e. [1] only obtains $T^{2/3}$. I don't think there has been work in this exact setting.

- However, contrary to the authors' claims, this is not the first paper in the area, as there have been papers on *group fairness* in bandits, which is one type of statistical parity, but not in the adversarial setting. The authors and reviewers seem to have completely missed this.
[1] Group Fairness in Bandit Arm Selection
[2] Group fairness in bandits with Biased Feedback
[3] Fairness-aware Bandit-based Recommendation
[4] Simultaneously Achieving Group Exposure Fairness and Within-Group Meritocracy in Stochastic Bandits
- The structure of the paper could be somewhat improved.

Most of the reviewers didn't seem to have a good grasp of the theory, so I take their evaluation with a grain of salt.